_Report_

EMBO
Molecular Medicine

# Clinical relevance of zebrafish for gene variants testing. Proof-of-principle with _SMN1_/SMA

Brett W Stringer [1], Yougang Zhang[1], Afsaneh Taghipour-Sheshdeh[1], Shuxiang Goh [2], Heike Kölbel[3], Michelle A Farrar [2,4], Brunhilde Wirth [5,6,7] & Jean Giacomotto [1,8,9,10]✉

## Abstract

Spinal muscular atrophy (SMA) results from _SMN1_ gene loss-of-function (LOF), with disease severity directly linked to the level of remaining SMN protein. Nusinersen, risdiplam, and onasemnogene abeparvovec are revolutionary treatments but should ideally be implemented before clinical symptoms appear. Because of this, prenatal and newborn screenings are increasingly used to identify common _SMN1_ variants and patients requiring therapy. However, for novel variants, clinicians lack robust analytic tools to predict pathogenicity before irreversible damage occurs. To address this gap, we deployed a zebrafish model presenting _smn1_-LOF, exhibiting progressive motor defects and death by only six days of age. We evaluated two _SMN1_-variants of uncertain significance (VUS) identified in newborn infants awaiting definite diagnosis and treatment recommendations. We demonstrated that while known pathogenic variants did not change the disease course, wild-type _SMN1_ and both infants variants rescued SMA hallmarks in zebrafish, demonstrating the relevance of this approach for VUS-testing within a crucial timeframe for patients. Extending the assay to known _SMN1_-hypomorphs showed partial rescue, weaker than wild-type or VUS, demonstrating that this approach can also discriminate partial-LOF effects. Both VUS were resolved to be non-pathogenic, and the therapeutic costs of >US$2 million per child were avoided. Beyond SMA, this study provides robust proof-of-principle that the zebrafish represents a powerful translational tool for VUS-analysis, and that such approaches should be considered in clinical settings for supporting diagnosis and treatment decisions.

**Keywords** Motor Neuron Disease; Diagnosis; Functional Test; Precision Medicine; Hypomorphic Variants; VUS Resolution
**Subject Category** Genetics, Gene Therapy & Genetic Disease

## Introduction

Spinal muscular atrophy (SMA) is a genetic disorder characterized by loss of motor neurons in the spinal cord and brain stem, leading to progressive muscle weakness and atrophy. It is caused by biallelic deletions or pathogenic variants of the survival motor neuron 1 (_SMN1_) gene, which results in a deficiency of the survival motor neuron (SMN) protein, which is crucial for motor neuron function and survival (Yeo et al, 2024). SMA severity is directly influenced by the _SMN2_ gene copy number, which is highly variable among individuals (Feldkotter et al, 2002). Without treatment, individuals with the most common form of the disorder, SMA type I, never gain the ability to sit or stand and usually die or require permanent ventilation within the first 2 years of life (Fig. 1A) (Mercuri et al, 2018; Wirth, 2021; Wirth et al, 2020; Yeo et al, 2024).

Until very recently, SMA was the most common inherited cause of infant mortality worldwide (Yeo et al, 2024). Thanks to intensive research efforts, three treatments have now been approved and are changing the lives of patients and their families. These treatments include the antisense oligonucleotide nusinersen, the small molecule splicing modifier risdiplam, and the gene therapy onasemnogene abeparvovec, all of which aim to increase SMN protein levels (Wirth, 2021; Wirth et al, 2020; Yeo et al, 2024).

However, to be optimally effective, these treatments should be implemented in pre-symptomatic patients, i.e., prior to the appearance of any major SMA signs or symptoms (Farrar et al, 2023; Finkel and Benatar, 2022; Kariyawasam et al, 2023; Tizzano and Zafeiriou, 2018). Indeed, when patients present with symptoms, irreversible motoneuron death has already occurred (Fig. 1A).

To address this dilemma, efforts are being made to diagnose SMA as early as possible, and many countries have now adopted prenatal and/or newborn screening to identify common pathogenic _SMN1_ mutations and patients to be treated. The majority (95%) of individuals with SMA are associated with biallelic _SMN1_ exon 7 deletions (Wirth et al, 2020; Yeo et al, 2024). Consequently, this is the target analyte in population screening programmes. However, in some cases, novel variants (variants of uncertain significance, VUS) are identified, and the question of their pathogenicity is thereby crucial.

[1]Institute for Biomedicine and Glycomics, Griffith University, Brisbane, QLD 4111, Australia. [2]Discipline of Paediatrics and Child Health, School of Clinical Medicine and Health, UNSW Medicine, UNSW Sydney, Sydney, NSW 2033, Australia. [3]Department of Pediatric Neurology, Centre for Neuromuscular Disorders, Centre for Translational Neuro- and Behavioral Sciences, University Hospital Essen, Essen, Germany. [4]Department of Neurology, Sydney Children's Hospital, Randwick, NSW 2031, Australia. [5]Institute of Human Genetics, University Hospital of Cologne, Cologne 50937, Germany. [6]Center for Molecular Medicine Cologne, University of Cologne, Cologne 50931, Germany. [7]Center for Rare Diseases, University Hospital of Cologne, Cologne 50937, Germany. [8]School of Environment and Science, Griffith University, Brisbane, QLD 4111, Australia. [9]Thompson Institute, National PTSD Research Centre, University of the Sunshine Coast, Birtinya, QLD 4575, Australia. [10]Queensland Brain Institute, The University of Queensland, Brisbane, QLD 4067, Australia. ✉E-mail: j.giacomotto@griffith.edu.au

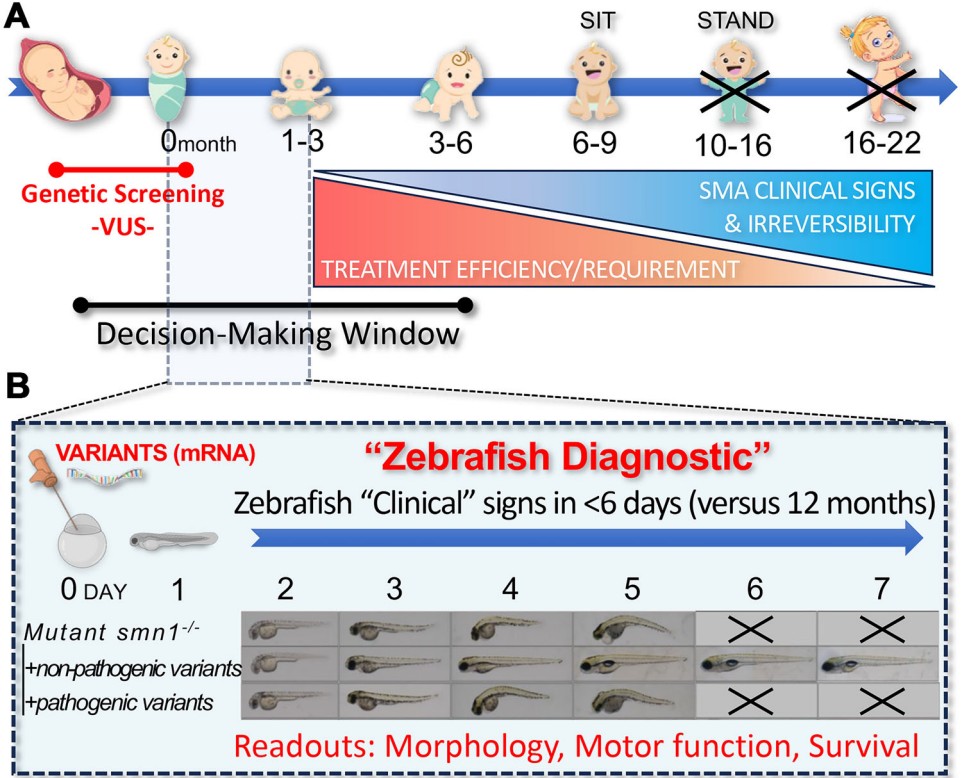

**Figure 1. Zebrafish in vivo functional assays can rapidly provide precious information to clinicians regarding the pathogenicity of variants of uncertain significance (VUS).**

(A) Schematic timeline of infant development highlighting the clinical/timeframe dilemma for detection and confirmation of *SMN1*-VUS pathogenicity for early-onset SMA (Type I/II), i.e., treatments need to be implemented before the first major clinical signs. Available treatments are expensive and potentially harmful if unnecessary, emphasizing the need for rapid and innovative VUS-testing approaches within a timeframe that aligns with disease progression. (B) Zebrafish functional/complementation assays can quickly provide valuable information on VUS-pathogenicity in less than 3 months, supporting clinicians in their decision process and in a clinically helpful timeframe. This rapid testing framework is applicable not only to spinal muscular atrophy (SMA) but also to a wide range of pediatric diseases, offering significant benefits in clinical practice.

Indeed, if pathogenic, definitive confirmation to enable treatment implementation before phenoconversion is important. On the contrary, confirming a variant is non-pathogenic will alleviate psychosocial distress and clinical surveillance (Kariyawasam et al, 2021). Unnecessary treatment implementation could also be harmful and significantly expensive; e.g., onasemnogene abeparvovec costs more than US$2 million for a single dose, and nusinersen more than US$4 million a decade (Belancic et al, 2025; Chan et al, 2023; Nuijten, 2022; Yeo et al, 2024).

Clinicians and patients would strongly benefit from the development of innovative methodology to help support VUS resolution. The zebrafish animal model holds tremendous promise in supporting clinical efforts to functionally characterize human disease-associated mutations (Adhish and Manjubala, 2023). Owing to its conserved neural circuitry, rapid development, and early spontaneous motor activity, zebrafish is particularly well suited for investigating neurological and neuromuscular disorders (Singh and Patten, 2022). Zebrafish have been extensively used to study SMA, and a variety of models exist—from maternal zygotic mutants lacking detectable SMN expression and used in this study (Boon et al, 2009; Hao le et al, 2013), to partial-LOF and tissue-specific models (Giacomotto et al, 2015; Guo et al, 2025; Laird et al,

2016). SMA in zebrafish triggers developmental regression, rapid motor decline, and early lethality, all taking place in the first week of life, making them an ideal model for rapid in vivo functional testing of *SMN1* variants. Here, we demonstrate that functional/complementation assays using zebrafish are a powerful complementary readout to support clinical decisions for the early-onset SMA forms, and, most importantly, fit within a timeframe that is congruent with pathophysiology (Fig. 1A,B).

## Results and discussion

Two cases, respectively carriers of *SMN1*-VUS c.861_864del (861VUS) and c.855_858del (855VUS) (Fig. 2) were identified at 2 weeks-of-age and brought to our attention by the Australian Functional Genomics Network (AFGN). Both *SMN1*-VUS were found in heterozygous but not homozygous state with an allele frequency of 0.000029 and 0.000008, respectively, in the Genome Aggregation Database (gnomAD™) v4, including exomes and genomes from 807,162 control individuals; functional prediction programs ranked them as likely pathogenic (https://gnomad.broadinstitute.org/). Each case/patient was simultaneously

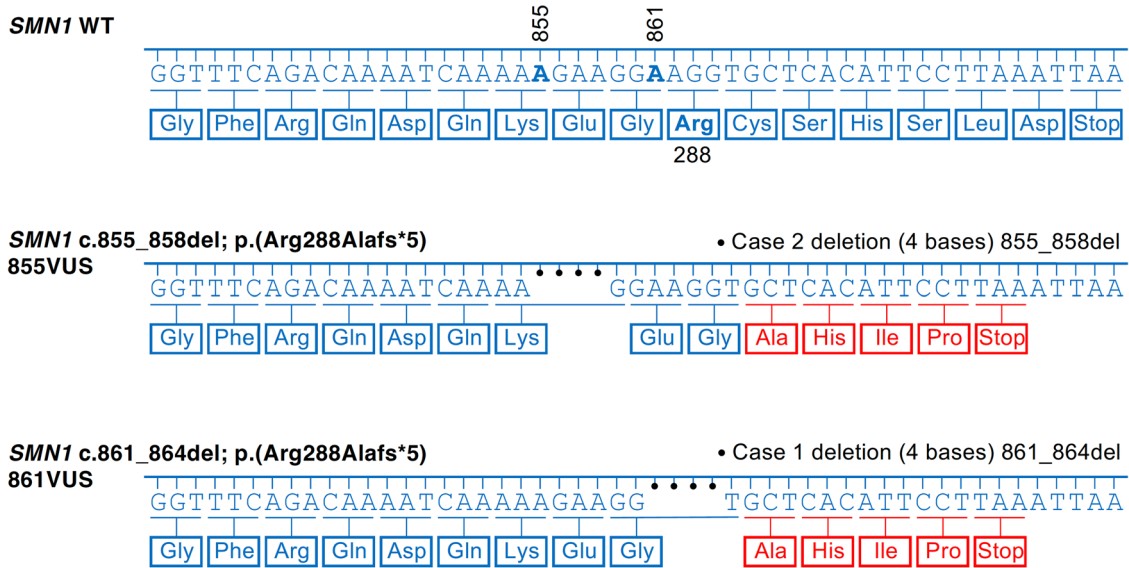

**Figure 2. Comparison of wild-type, 855VUS, and 861VUS *SMN1* nucleotide and amino acid sequences.**

The three panels show the nucleotide sequence of the coding strand of the wild-type (WT), 855VUS and 861VUS *SMN1* alleles, from the first nucleotide of the final coding exon to the position of the stop codon. The 3′ amino acid sequence of the wild-type SMN protein is shown in blue, with changes associated with each VUS in red. The location of coding sequence nucleotides 855 and 861 and amino acid 288 are highlighted in the wild-type strand/sequence, and deletions are marked with dots.

a carrier of a common pathogenic deletion of *SMN1* exon 7, putting them at risk of developing SMA. Moreover, one newborn failed to carry any *SMN2* copy, the other carried one *SMN2* copy, suggesting the potential to develop a severe SMA type. However, no clinical symptoms were present at the time of discussion, and the determination of the possible pathogenicity of each VUS was important to support either a decision to implement treatment or to possibly delay it until the emergence of the first subclinical electrophysiological SMA hallmarks/diagnosis. To address the potential pathogenicity of these VUS, we deployed a zebrafish SMA model deficient in SMN function and tested independently whether each variant could successfully complement the lack of SMN function and associated phenotypes (Fig. 1B).

## Patients' VUS rescued the morphological defects of the SMA zebrafish

To determine whether the *SMN1* variants were pathogenic, we investigated whether in vitro-transcribed mRNAs from 861VUS and 855VUS, injected into single-cell *smn*$^{-/-}$ zebrafish embryos, successfully rescued the previously described zebrafish SMA hallmarks, including morphology, swimming ability, and survival (Fig. 3; Appendix Figs. S1–S4) (Giacomotto et al, 2015; Hao le et al, 2013; Laird et al, 2016). To evaluate the efficiency of our approach, we used human wild-type *SMN1* mRNA ("WT"—positive control) that we previously demonstrated to be efficient in rescuing zebrafish SMA phenotypes (Giacomotto et al, 2015; Hao le et al, 2013; Laird et al, 2016). For robustness, we tested a series of additional controls, including (i) mock-injected *smn*$^{-/-}$ ("*smn*$^{-/-}$"— negative control), (ii) stable transgenics expressing human wt-*SMN1* mRNA ubiquitously ("*Tg(SMN1)*"—positive control) (Hao le et al, 2013), (iii) known pathogenic human *SMN1* mRNA variant (*SMN1* c.549del) ("path"—negative control) and (iv) non-

pathogenic human *SMN1* mRNA variant (*SMN1* c.462 A > G) ("non-path"—positive control).

As previously shown, and replicated here, absence of Smn/SMN protein in the *smn*$^{-/-}$ SMA zebrafish model triggered morphological malformations from 3 days post fertilization (dpf), with larvae being smaller in size and presenting body curvature as well as small eyes (Fig. 3A) (Hao le et al, 2013). These hallmarks became more prominent at 4 and 5 dpf. Pericardial and cerebral edema appeared from 4 dpf. Finally, 100% of larvae died between 5 and 6 dpf, as identified by the absence of heartbeats. None of the negative controls—mock-injections and pathogenic variant injections (path) —modified these disease hallmarks, with all embryos and larvae exhibiting the same malformations and with similar dynamics/ progression (Fig. 3A). In contrast, and confirming the robustness and relevance of our methodology, mRNAs from both wt-*SMN1* (WT) and a non-pathogenic *SMN1* variant (non-path) rescued these phenotypes. Importantly, mRNAs from both 861VUS and 855VUS also rescued these hallmarks, with all embryos and larvae having indistinguishable development/morphology when compared to all positive controls (WT, Tg(SMN1), non-path). This was clearly apparent from 3 dpf and persisted to 10 dpf, when the experiment was ended.

Together, these data demonstrate that mRNA from both *SMN1* VUS rescued the morphological phenotype of *smn*$^{-/-}$ SMA zebrafish as effectively as wt-*SMN1*, suggesting that each VUS properly translates into a functional SMN protein.

## Patients' VUS improved motor functions of SMA zebrafish as efficiently as wt-*SMN1*

To further investigate the function/pathogenicity of the identified VUS, we conducted functional assays assessing motor function recovery following SMN complementation. Swimming behavior

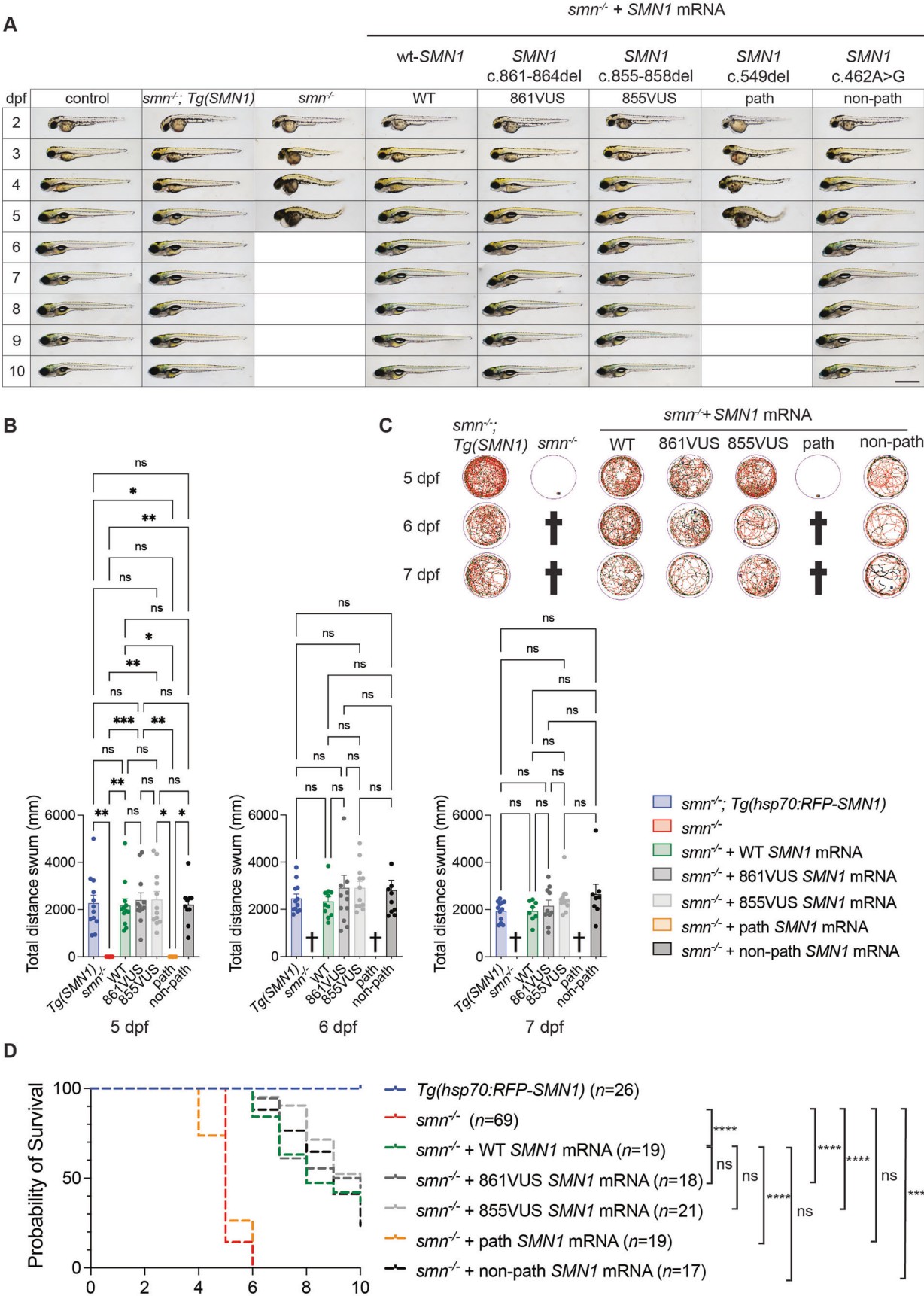

**Figure 3.   861VUS and 855VUS mRNA injections rescue SMA *smn*⁻/⁻ zebrafish disease hallmarks.**

(A) *smn*⁻/⁻ zebrafish (column 3) develop morphological malformations from 3 dpf, being smaller with progressive body curvature and small eyes; for comparison, see control zebrafish (column 1). From 4 dpf, *smn*⁻/⁻ animals develop progressive pericardial and cerebral edema. Animals died between 4 and 6 dpf, as determined by the absence of a heartbeat. mRNA from the known pathogenic *SMN1* variant, *SMN1* c.549del (path—negative control, column 7), as expected, failed to rescue this phenotype (did not prevent the appearance of these malformations). In contrast, positive controls: (i) ubiquitous transgenic expression of wt-*SMN1* (*Tg(SMN1)*, column 2), (ii) injected wt-*SMN1* mRNA (WT, column 4), (iii) injected non-pathogenic variant c.462 A > G (non-path, column 8), and importantly (iv) injected c.861_864del *SMN1* mRNA (861VUS, column 5) and injected c.855_858del *SMN1* mRNA (861VUS, column 6) all rescued these morphological traits, with no distinguishable difference between the animals. These results robustly demonstrate that both the 861VUS and 855VUS produce functional SMN protein. Empty boxes indicate 100% batch mortality. Scale bar, 500 μm. (B) SMA *smn*⁻/⁻ zebrafish present dramatic motor function loss at 5 dpf and died by 6 dpf. As previously demonstrated (Giacomotto et al, 2015), human wt-*SMN1* transgenic ubiquitous expression (*Tg(SMN1)*) or injected wt-*SMN1* mRNA (WT) rescue this motor function loss. Control *smn*⁻/⁻;*Tg(SMN1)* were heat-shocked once at 24 hpf to produce SMN. Similarly, the non-pathogenic *SMN1* variant (non-path) and both the 861VUS and 855VUS efficiently restore motor function with no detectable significant difference. On the contrary, confirming the robustness of the approach, mRNA from the pathogenic *SMN1* variant (path) failed to improve motor function with no difference from the negative control (*smn*⁻/⁻). Graphs represent comparisons of the total distance swum by each cohort of fish over 24 min. Each graph is representative of an experiment performed three times (Appendix Figs. S5–S4). Each data point represents one fish. Error bars represent the standard error of the mean. Statistical significance was evaluated using the Kruskal–Wallis test with Dunn's correction for multiple comparisons. ****$P < 0.0001$; ***$P < 0.001$; **$P < 0.01$; *$P < 0.05$; ns, not significant. Exact $P$ values are shown tabulated in Appendix Fig. S2. Source data are available online for this Figure. (C) Representative swimming tracks at 5, 6, and 7 dpf. Cross symbols indicate 100% mortality. (D) SMA *smn*⁻/⁻ zebrafish (*smn*⁻/⁻, red dashed line) had a median survival of 5 days. Injected mRNA from the known pathogenic *SMN1* variant (path, orange dashed line) did not have any effect on survival. In contrast, demonstrating restoration of Smn/SMN function, injected mRNA from 861VUS (dark gray dashed line) and 855VUS (light gray dashed line) extended the survival of *smn*⁻/⁻ larvae similar to wt-mRNA (WT, green dashed line) and non-pathogenic variant (non-path, black dashed line) animals. ****$P < 0.0001$; ns, not significant. Statistical significance was evaluated using the log-rank (Mantel–Cox) test. Exact $P$ values and median survival data are shown tabulated in Appendix Fig. S2. Source data are available online for this figure.

and ability of zebrafish larvae were analyzed using automatic tracking at 5, 6, and 7 dpf as per the method section. At 5 dpf, mock-injected SMA *smn*⁻/⁻ larvae (*smn*⁻/⁻) exhibited dramatic motor function loss, as demonstrated by a near-absent spontaneous swimming pattern and response to stimulation (Fig. 3B,C). As expected, injections of the pathogenic variant (path) failed to rescue these motor function losses and were not significantly different from mock-injected SMA *smn*⁻/⁻ control animals. In contrast, injection of either wt-*SMN1* (WT), non-pathogenic variant (non-path), and both patients' 861VUS and 855VUS, robustly restored motor function as demonstrated by significantly longer distance swum with no difference to transgenic control (Fig. 3B,C; Appendix Figs. S2–S4A,B). To confirm the robustness and reproducibility of these findings, the assays were independently repeated three times. As shown in Appendix Figs. S2–S4, all replicates yielded consistent outcomes. Power analysis using the triplicate datasets at 5 dpf revealed a very large effect size (Cohen's d ≈ 3.3; ANOVA f ≈ 1.65, one-way ANOVA, GPower 3.1) between positive and negative controls, indicating that as few as three larvae per group were sufficient to discriminate pathogenic from non-pathogenic variants at 90% power ($\alpha = 0.05$), underscoring the high reliability and sensitivity of this functional assay.

It is noteworthy that at 6 dpf, all negative control larvae were dead (mock-injection and pathogenic variant), demonstrating proper progression of the disease and execution of the experiments. In contrast, *smn*⁻/⁻ larvae in the remaining cohorts remained alive and healthy, indistinguishable from injected positive controls, and continued to swim without significant difference at both 6 dpf and 7 dpf (Fig. 3B,C; Appendix Figs. S2–S4A,B).

Together, these data demonstrate that both patients' VUS mRNAs rescue motor function of SMA *smn*⁻/⁻ zebrafish as efficiently as wt-*SMN1* mRNA or non-pathogenic *SMN1* variant mRNA.

### Patients' VUS increased the survival of SMA zebrafish as efficiently as wt-*SMN1*

We complemented our readout/assays by investigating the effect of both 861VUS and 855VUS on survival. In the absence of Smn/SMN

protein, SMA *smn*⁻/⁻ zebrafish (*smn*⁻/⁻) have a highly replicable median survival of 5–6 days post-fertilization (Fig. 3D; Appendix Figs. S2–S4C,D) (Hao le et al, 2013). While the addition/injection of pathogenic *SMN1* variant mRNA (path) failed to modify this mortality in all experiments conducted, wt-*SMN1* (WT), non-pathogenic *SMN1* (non-path) and patients' VUS (861VUS and 855VUS) extended the median survival to over 8 days. These data robustly support that the patients' VUS retain biological function and do not differ statistically in our assay from the positive controls based on wt-*SMN1* and non-pathogenic *SMN1* variants. Experiments were stopped at 10 dpf as injected mRNAs have a time-restricted efficacy, being degraded in the first week after injection (Bondue et al, 2023; Fink et al, 2006).

Taken together, these data show that both patients' VUS rescued all tested SMA hallmarks in the zebrafish (morphology, motor function, and survival), clearly indicating that the protein encoded by these novel variants retains significant biological function.

Fully confirming the relevance of the presented methodology, the patients were around 6 months of age at the time of experiment completion and were still asymptomatic. After review and consultation with clinicians, a decision was made not to implement treatment and wait until the patients were 12 months old to write this manuscript, at which time both infants remained asymptomatic, exhibiting normal motor, neurological and electrophysiology examination. Both patients achieved all motor milestones at expected timepoints, with patient 1 attaining independent walking at 14 months of age.

### Potential for the detection of hypomorphic variants and late-onset SMA

This approach was designed to offer a rapid functional assay to assist clinicians in confirming or excluding the risk of SMA type I in newborn carriers of *SMN1*-VUS, where timely intervention is critical to prevent irreversible damage. While type I is the most severe and diagnostically challenging form of SMA, it would be interesting to know if this methodology could also be informative for intermediate and mild forms, such as SMA types II and III. To

**A**

| | Variant | | SMA type | SMN2 | PMID |
|---|---|---|---|---|---|
| 1 | *SMN1* c.861-864del (p.Arg288Alafs*5) | 861VUS | this work | this work | this work |
| 2 | *SMN1* c.855-858del (p.Arg288Alafs*5) | 855VUS | this work | this work | this work |
| 3 | *SMN1* c.549del (p.Lys184fs) | path 1 | I | *n*=1 | 34602496 |
| 4 | *SMN1* c.462A>G (p.Gln154=) | non-path 1 | benign | *n*=1 | ClinVar |
| 5 | *SMN1* c.43C>T (p.Gln15*) | path 2 | I | *n*=1 | 10205265 |
| 6 | *SMN1* c.84C>T (p.Ser28=) | non-path 2 | benign | *n*=1 | ClinVar |
| 7 | *SMN1* c.5C>G (p.Ala2Gly) | A2G | II | *n*=1 | 39905579 |
| 8 | *SMN1* c.5C>T (p.Ala2Val) | A2V | III | *n*=1 | 24359787 |
| 9 | *SMN1* c.131A>T (p.Asp44Val) | D44V | III | *n*=1 | 15580564 |
| 10 | *SMN1* c.734C>T (p.Pro245Leu) | P245L | III | *n*=1 | 10732817 |
| 11 | *SMN1* c.785G>T (p.Ser262Ile) | S262I | III | *n*=1 | 10699387 |
| 12 | *SMN1* c.821C>T (p.Thr274Ile) | T274I | II | *n*=1 | 39905579 |

**B**

*smn⁻/⁻ + SMN1* mRNA

| dpf | control | *smn⁻/⁻* | *smn⁻/⁻; Tg(SMN1)* | wt-*SMN1* WT | ① *SMN1* c.861-864del 861VUS | ② *SMN1* c.855-858del 855VUS | ③ *SMN1* c.549del path 1 | ④ *SMN1* c.462A>G non-path 1 |

*smn⁻/⁻ + SMN1* mRNA

| dpf | ⑤ *SMN1* c.43C>T (p.Q15*) path 2 | ⑥ *SMN1* c.84C>T (pS28=) non-path 2 | ⑦ *SMN1* c.5C>G A2G (type II) | ⑧ *SMN1* c.5C>T A2V (type III) | ⑨ *SMN1* c.131A>T D44V (type III) | ⑩ *SMN1* c.734C>T P245L (type III) | ⑪ *SMN1* c.785G>T S262I (type III) | ⑫ *SMN1* c.821C>T S274I (type II) |

◄ **Figure 4.  mRNA injections of *SMN1* hypomorphic variants associated with SMA type II and III only partially rescue the phenotype of *smn*⁻/⁻ zebrafish, supporting the method's ability to detect hypomorphic variants.**

(A) Details of the *SMN1* variants selected for further analysis. Eight additional variants were evaluated: two associated with SMA type II, four with SMA type III, one additional known pathogenic variant (path 2), and one additional known non-pathogenic variant (non-path 2). All variants were selected to be specifically associated with patients carrying a single *SMN2* copy, providing a uniform genetic background and avoiding confounding effects related to *SMN2* dosage. (B) Morphological malformations consistent with SMA-hallmarks were observed in *smn*⁻/⁻ embryos injected with all six *SMN1* type II/III variants, although these appeared later or in milder forms than in mock-injected negative controls or embryos injected with SMA Type I pathogenic variants *SMN1* c.549del (path 1) and *SMN1* c43C>T (p.Gln15*) (path 2). In contrast, mRNA encoding non-pathogenic *SMN1* variants, *SMN1* c.462 A > G (non-path 1) and *SMN1* c.84 C > T (non-path 2) as well as the two VUS (861VUS and 855VUS), fully rescued the *smn*⁻/⁻ phenotype, showing no difference from the wt-*SMN1* mRNA positive control. These results provide further evidence of functional SMN protein expression and confirm the non-pathogenic nature of the tested VUS. Empty boxes indicate all larvae are dead. Scale bar, 500 μm. Source data are available online for this figure.

investigate this and at the same time better addressing the standard guidelines in VUS resolution (Brnich et al, 2019), we further selected several *SMN1* variants known to be associated with SMA type II or III, along with extra pathogenic and non-pathogenic controls and compared these new conditions to all previously tested variants, via triplicate experiments (Figs. 4, 5 and EV1; Appendix Figs. S5 and S6).

All hypomorphic variants significantly improved the morphology and survival of SMN-deficient *smn*⁻/⁻ animals, but none reached the rescue efficiency of wt, non-pathogenic controls or VUS855 and VUS861 mRNAs (Figs. 4A,B; 5A,B; Appendix Figs. S5 and S6A,B). These results robustly confirm the non-pathogenicity of both VUS and demonstrate that our assay can detect partial-LOF (hypomorphic) variants. While morphology and survival endpoints clearly distinguished hypomorphs from both negative and positive controls as well as from the VUS, motor-function assays failed to segregate them from the negative controls, with swimming performance not significantly differing from mock-injected or pathogenic controls (path 1 and path 2) (Fig. EV1A–C). While these motor-function assays could be further refined in the future to enhance hypomorph discrimination, in the present study, they nonetheless reinforce the non-pathogenic nature of VUS861 and VUS855.

## Interpretation of findings

In recent years, the introduction of nusinersen, risdiplam, and onasemnogene abeparvovec has dramatically transformed the landscape of SMA, which was one of the most common inherited causes of infant mortality worldwide. For early-onset/severe forms of SMA, these treatments are most effective when administered before the onset of clinical symptoms (Fig. 1) (Balaji et al, 2023; Yeo et al, 2024). This critical time window demands early and accurate diagnosis of SMA, which has been facilitated by widespread prenatal and newborn screening programs for the detection of common/known mutations. However, the interpretation of novel VUS remains a formidable challenge in the clinical setting. Our study demonstrates the potential of zebrafish as a powerful translational model for the functional characterization of *SMN1* VUS, providing essential data to support both diagnosis and clinical decision-making.

We investigated two *SMN1* VUS identified in neonates through newborn screening. These two different variants, c.861_864del (861VUS) and c.855_858del (855VUS), are predicted to encode the same truncated SMN protein. Our functional assays demonstrated that, when injected in the early embryo, mRNAs from 861VUS and 855VUS could efficiently rescue the morphological, motor function,

and survival defects in SMA *smn*⁻/⁻ zebrafish larvae as effectively as human wt-*SMN1* mRNA. These findings suggest that the proteins encoded by these variants retain sufficient biological function to compensate for the loss of endogenous SMN protein, thus providing crucial evidence for their non-pathogenicity regarding severe forms of SMA. Extending this comparative approach to known hypomorphic *SMN1* variants further confirmed that both VUS behave as wild-type (Figs. 4, 5, and EV1; Appendix Figs. S5 and S6).

Given the pressing need to reach a timely clinical decision, and because (i) both VUS proved to be non-pathogenic in the zebrafish assays, (ii) the patients were still asymptomatic at 6 months of age and (iii) case 2 carries no *SMN2* copy (associated with early pathogenicity or lethality), the clinical team decided not to initiate therapy but instead to continue monitoring until 12 months of age before finalizing this report. At that time, both infants were within normal growth parameters and met expected motor milestones. At the time of manuscript submission, both patients were 14 months of age and remained asymptomatic. The initiation of drug therapy with potential side effects and the physical and psychological stress for the child and parents could be avoided. In addition, the public health system did not bear the therapy costs of around US$2 million (or more) per child.

Our study also strongly supports variant reclassification as benign. Considering the novelty of this approach and the complexity of SMA into late-onset forms, the group decided to continue monitoring the infants and complement our zebrafish rapid assay with long-term transgenic complementation experiments. Formal reclassification will be initiated at the end of this follow-up clinical study and transgenesis approach.

As a guide, the assessment of these *SMN1* variants for urgent clinical decision-making was completed in 18 working days (Fig. 1). This did not include the initial synthesis of coding DNA sequence (cds) by an external supplier, a time constraint that in the future should add no more than approximately two weeks to the pipeline. Functional evaluation of *SMN1* RNA in *smn*⁻/⁻ zebrafish embryos and larvae was the shortest step of the process, and clear evidence of SMN function was apparent in as little as six days. In conclusion, this study demonstrates that zebrafish is a robust and inexpensive tool to assess *SMN1*-VUS and should be considered in clinical settings to support the diagnosis of severe SMA forms and the need for treatment.

Importantly, as newborn screening (NBS) for SMA is expanding, additional babies are anticipated to be identified with novel variants. As this will cause intense distress for families, rapid and accurate VUS resolution is therefore highly impactful and will help in guiding a definitive clinical pathway, either treatment or

A

| Variant | | SMA type | SMN2 | Median survival (dpf) | vs smn⁻/⁻ | vs WT |
|---|---|---|---|---|---|---|
| ① *SMN1* c.861-864del (p.Arg288Alafs*5) | 861VUS | | | 8 | **** *P*<0.0001 | *ns, P*=0.4107 |
| ② *SMN1* c.855-858del (p.Arg288Alafs*5) | 855VUS | | | 9 | **** *P*<0.0001 | *ns, P*=0.4076 |
| ③ *SMN1* c.549del (p.Lys184fs) | path 1 | I | *n*=1 | 5 | *ns, P*=0.5982 | **** *P*<0.0001 |
| ④ *SMN1* c.43C>T (p.Gln15*) | path 2 | I | *n*=1 | 5 | *ns, P*=0.0966 | **** *P*<0.0001 |
| ⑤ *SMN1* c.462A>G (p.Gln154=) | non-path 1 | benign | *n*=1 | 9 | **** *P*<0.0001 | *ns, P*=0.5323 |
| ⑥ *SMN1* c.84C>T (p.Ser28=) | non-path 2 | benign | *n*=1 | 9 | **** *P*<0.0001 | *ns, P*=0.2866 |
| ⑦ *SMN1* c.5C>G (p.Ala2Gly) | A2G | II | *n*=1 | 5 | ** *P*=0.0046 | **** *P*<0.0001 |
| ⑧ *SMN1* c.5C>T (p.Ala2Val) | A2V | III | *n*=1 | 6 | **** *P*<0.0001 | **** *P*<0.0001 |
| ⑨ *SMN1* c.131A>T (p.Asp44Val) | D44V | III | *n*=1 | 6 | **** *P*<0.0001 | **** *P*<0.0001 |
| ⑩ *SMN1* c.734C>T (p.Pro245Leu) | P245L | III | *n*=1 | 6 | **** *P*<0.0001 | **** *P*<0.0001 |
| ⑪ *SMN1* c.785G>T (p.Ser262Ile) | S262I | III | *n*=1 | 6 | **** *P*<0.0001 | **** *P*=0.0001 |
| ⑫ *SMN1* c.821C>T (p.Thr274Ile) | T274I | II | *n*=1 | 6 | **** *P*<0.0001 | **** *P*<0.0001 |

B

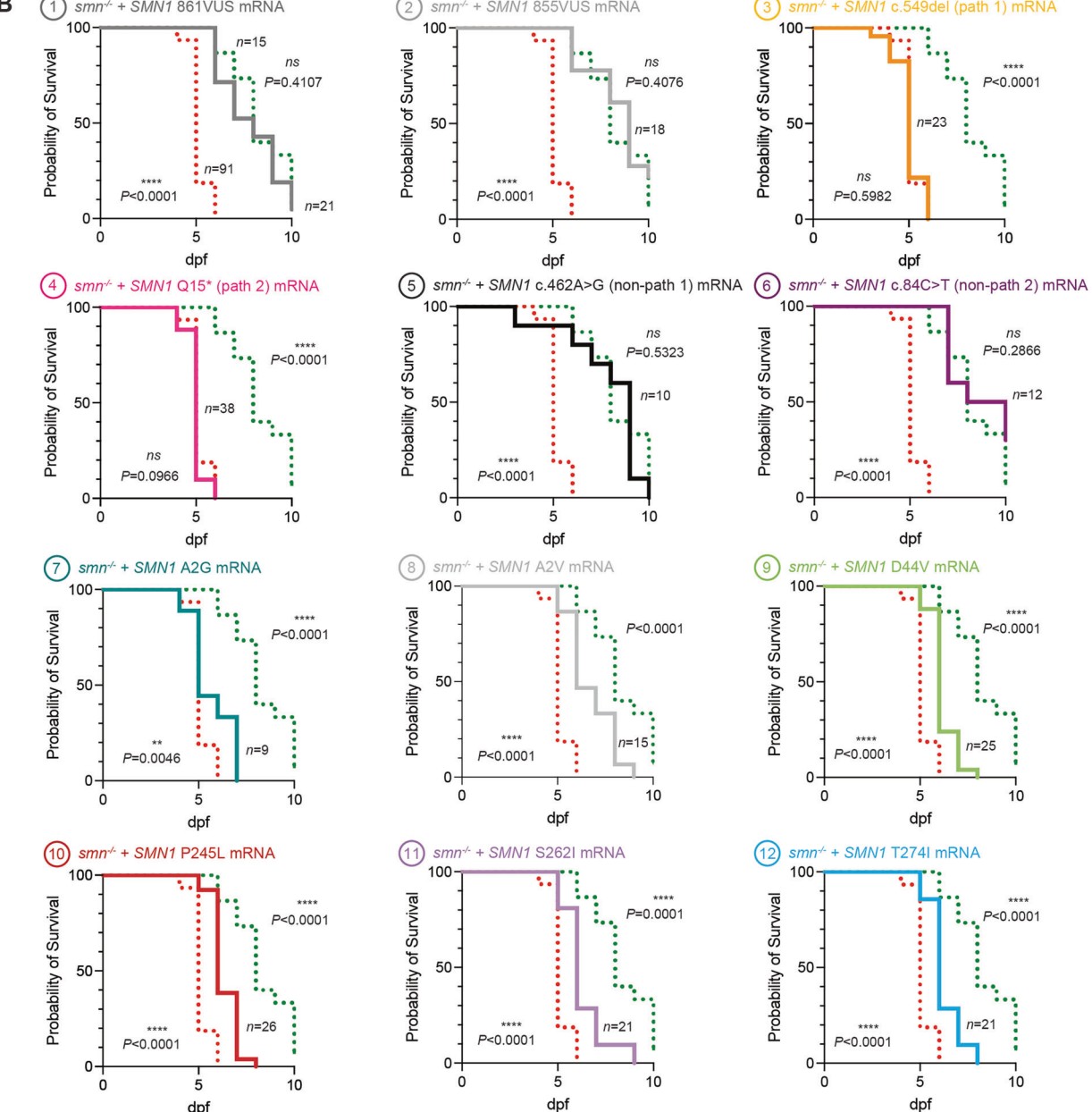

**Figure 5.  mRNA injections of *SMN1* hypomorphic variants extend the survival of SMN-deficient zebrafish, but only partially compared with wild-type or non-pathogenic controls.**

(A) Median survival of $smn^{-/-}$ larvae injected with each of the *SMN1* variants associated with SMA type II or III, was compared with both VUS and with known pathogenic and non-pathogenic *SMN1* variants. Kaplan–Meier analysis was used to determine median survival. Statistical significance, compared to $smn^{-/-}$ mock-injected control fish ($smn^{-/-}$) and wt-*SMN1* mRNA-injected animals (WT), was determined by log-rank (Mantel–Cox) test. ****$P < 0.0001$; *$P < 0.05$; ns, not significant. Exact $P$ values are also shown. Data are representative of one experiment performed three times (see Appendix Figs. S5 and S6 for additional replicates). (B) Survival curves for each of the tested *SMN1* variants (solid colored line) shown relative to those for $smn^{-/-}$ mock-injected control fish (dotted red line) and wt-*SMN1* mRNA-injected animals (dotted green line). $P$ values are as in (A). All tested missense variants, associated with mild forms of the disease, significantly extended the survival of SMN-deficient $smn^{-/-}$ animals, though none matched the rescue efficiency of wt or VUS mRNA, suggesting that the presented methodology would also be useful to detect hypomorphs. These findings, combined with the data presented in Figs. 3 and 4, support that the tested 861VUS and 855VUS variants are neither pathogenic nor hypomorphic. Source data are available online for this figure.

reduction of surveillance, stress, and uncertainty for families and health services.

## Limitations

The presented zebrafish complementation assays provide a rapid and practical functional approach to aid in VUS resolution and support timely treatment decisions for SMA type I, where early intervention is critical. For the less urgent support in the diagnosis of SMA type II and III and for formal VUS reclassification, this methodology may warrant further investigation. While our preliminary results with hypomorphic variants suggest that the presented assays could ultimately be used for formal variant classification, caution is warranted as these results currently exceed expectations. To the best of our knowledge, no studies so far have successfully visualized *SMN1* hypomorphic pathogenicity in the absence of *SMN2*, nor have pathogenic variants been identified in patients with no copy of *SMN2*, suggesting that the presence of at least one copy of the *SMN2* gene is essential for a *SMN1* missense variant to act as a hypomorph (Blatnik et al, 2020; Iyer et al, 2018). Our system, which evaluates phenotypes within the first week of life, may be sensitive enough to detect such hypomorphic effects even in the absence of endogenous SMN protein. Alternatively, some residual maternal SMN deposition might be present at birth in $smn^{-/-}$ animals, though below the limits of detection by western blot, both in the study by Hao et al (Hao le et al, 2013) and in this study (Appendix Figs. S7 and S8). In short, while our assay is a valuable and timely tool for supporting SMA type I diagnostics, its application to predicting late-onset type II/III forms should, for now, be approached with caution. For the latter, where immediate intervention is less urgent, we recommend combining this zebrafish-based assay with complementary models, such as rodents, cell lines, and the methodology described by Blatnik et al (Blatnik et al, 2020). Finally, because the presented transient mRNA assays cannot capture effects on splicing or transcription, zebrafish results should be complemented—whenever possible—with patient biopsies to evaluate possible alternative splicing and expression levels in patients.

## Implications for clinical practice and application to other diseases

Going beyond SMA, our study provides robust proof-of-principle that zebrafish can serve as a powerful tool for VUS resolution. To the best of our knowledge, this represents the first demonstration of

their successful translational use within a clinically meaningful timeframe. Indeed, our study highlights the clinical relevance of zebrafish as a valuable model for gene variants (VUS) testing, not only for SMA, but for human disease in general. With exome and whole genome sequencing becoming more commonplace, we are seeing an explosion of new gene variants associated with a diversity of disorders. The determination of their functional consequences is posing a significant challenge, and this problem will continue to grow with the development and need for precision/personalized medicine. The development of robust tools for VUS resolution is becoming critical. Our work shows that the zebrafish can help address this translational need.

## Methods

### Reagents and tools table

| Reagent/resource | Reference or source | Identifier or catalog number |
|---|---|---|
| **Experimental models** | | |
| *Danio rerio* | Boon et al (2009) Hao et al (2013) | $smn^{-/-}$; Tg(hs:RFP-SMN^{+/−}) |
| *Danio rerio* | Punnamoottil et al (2015) | *MN:GFP* |
| **Recombinant DNA** | | |
| pT3TS-Dest_R1-R3 | Tromp et al, 2023 | Cat #140878 |
| p3E-polyA | Kwan et al (2007) | N/A |
| 249-pME-hsa_SMN1-wt | This study | Appendix Table S1 |
| 250-pME-c.861_864_SMN1 | This study | Appendix Table S1 |
| 252-pME-c.855_858_SMN1 | This study | Appendix Table S1 |
| 253-pME-c.549del_SMN1 | This study | Appendix Table S1 |
| 254-pME-c.462 A > G_SMN1 | This study | Appendix Table S1 |
| 260-pT3-hsa_SMN1-WT_polyA | This study | Appendix Table S1 |
| 261-pT3-c.861_864-hsa_SMN1_polyA | This study | Appendix Table S1 |
| 263-pT3-c.855_858_hsa_SMN1_polyA | This study | Appendix Table S1 |

| Reagent/resource | Reference or source | Identifier or catalog number |
|---|---|---|
| 264-pT3-c.549del_hsa_SMN1_polyA | This study | Appendix Table S1 |
| 265-pT3-c.462 A > G_hsa_SMN1_polyA | This study | Appendix Table S1 |
| 279-hsa-SMN1c.5 C > G(p.Ala2Gly) | This study | Appendix Table S1 |
| 280-hsa-SMN1c.5 C > T(pAla2Val) | This study | Appendix Table S1 |
| 281-hsa-SMN1c.84 C > T(p.Ser28 = ) | This study | Appendix Table S1 |
| 282-hsa-SMN1c.131 A > T(p.Asp44Val) | This study | Appendix Table S1 |
| 283-hsa-SMN1c.734 C > T(p.Pro245Leu) | This study | Appendix Table S1 |
| 284-hsa-SMN1c.785 G > T(p.Ser262Ile) | This study | Appendix Table S1 |
| 285-hsa-SMN1c.821 C > T(p.Thr274Ile) | This study | Appendix Table S1 |
| 287-pT3-hsa_SMN1_A2G | This study | Appendix Table S1 |
| 288-pT3-hsa_SMN1_A2V | This study | Appendix Table S1 |
| 289-pT3-hsa_SMN1_S28= | This study | Appendix Table S1 |
| 290-pT3-hsa_SMN1_D44V | This study | Appendix Table S1 |
| 291-pT3-hsa_SMN1_P245L | This study | Appendix Table S1 |
| 292-pT3-hsa_SMN1_S262I | This study | Appendix Table S1 |
| 293-pT3-hsa_SMN1_T274I | This study | Appendix Table S1 |
| **Antibodies** | | |
| Mouse anti-SMN | Glenn E. Morris, Developmental Studies Hybridoma Bank | MANSMN12, 2E6 |
| Mouse anti-β-actin | Sigma-Aldrich | A5441-.2 ML |
| Horse anti-mouse IgG HRP-linked secondary antibody | Cell Signaling | 7076 s |
| **Oligonucleotides and other sequence-based reagents** | | |
| qPCR primers | This study | Appendix Table S2 |
| **Chemicals, enzymes, and other reagents** | | |
| One Shot™ TOP10 Chemically Competent *E. coli* | ThermoFisher Scientific | Cat #C404010 |
| Luria Bertani Broth (Miller) | Oxoid | Cat #CM0996 |
| Agar Bacteriological (Agar No.1) | Oxoid | Cat #LP0011B |
| Ampicillin sodium salt | Sigma-Aldrich | Cat #A9518-5G |
| Kanamycin Sulfate | Gibco | Cat #11815-024 |
| Gateway™ LR Clonase™ II Enzyme mix | ThermoFisher Scientific | Cat #1791100 |
| GeneJET Plasmid Miniprep Kit | ThermoFisher Scientific | Cat #K0503 |

| Reagent/resource | Reference or source | Identifier or catalog number |
|---|---|---|
| *Kpn*I-HF | New England BioLabs | Cat #R3142S |
| PureLink™ Quick Gel Extraction and PCR Purification Combo Kit | ThermoFisher Scientific | Cat #K220001 |
| mMESSAGE mMACHINE™ T3 Transcription Kit | ThermoFisher Scientific | Cat # AM1348 |
| MEGAclear™ Transcription Clean-Up Kit | ThermoFisher Scientific | Cat #AM1908 |
| TRIzol™ Reagent | ThermoFisher Scientific | Cat # 15596026 |
| Directzol RNA MicroPrep Kit | Zymo Research | Cat #R2062 |
| QuantiTect Reverse Transcription Kit | QIAGEN | Cat #205311 |
| QIAGEN Quantitect SYBR Green PCR Kit | QIAGEN | Cat #204163 |
| Pierce™ BCA Protein Assay Kits | Thermo Scientific™ | Cat #23225 |
| 0.45 μm hydrophobic PVDF membrane (Immobilon®-P) | Merck Millipore | Cat #IPVH00010 |
| **Software** | | |
| GraphPad Prism | GraphPad Software | Version 10.2.3 (347), April 21, 2024 |
| **Other** | | |

## Human ethics

Human ethics approval for this work was granted by the Sydney Children's Hospitals Network Human Research Ethics Committee, approval number 2023/ETH01937. Informed consent was obtained from the parents of both subjects, and experiments conformed to the principles set out in the WMA Declaration of Helsinki and the Department of Health and Human Services Belmont Report.

## Cases/patients

Case 1 was a 12-day-old asymptomatic male infant with no reported family history of SMA, identified through the New South Wales Newborn Screening Programme (Kariyawasam et al, 2020). The initial qPCR PerkinElmer NBS assay identified 0 *SMN1* and second-tier ddPCR assays identified 1 *SMN1* and 1 *SMN2* copy. This result was interpreted as an uncertain newborn screening (NBS) for SMA, and the family was contacted for further testing. Diagnostic testing, including *SMN* sequencing, determined that the proband was a compound heterozygote. The infant had a common *SMN1* exon 7 large deletion and an intragenic 4-base pair deletion in exon 7, NM_000344.3, NP_000335.1: c.861_864del (p.Arg288Alafs*5) (Fig. 2). The latter mutation was located under the primer used in the first-tier NBS assay but was not in the region detected by the ddPCR assay. This intragenic *SMN1* mutation was classified by the testing laboratory and prediction tools as a variant of uncertain significance, VUS-3A, and is referred to as 861VUS in this manuscript. This 861VUS (*SMN1*c.861_864del (p.Arg288Alafs*5)) has been reported 47 times in the gnomAD database from

807,162 exomes and genomes (gnomAD™ v4.1.0, June 2025). The VUS3A classification was primarily based on its low frequency in gnomAD controls, and the fact that it is not located within a functional domain, mutational hotspot, or highly conserved region, nor in a region typically associated with nonsense-mediated decay.

Case 2 was a 7-day-old asymptomatic female infant with no reported family history of SMA, identified with 0 *SMN1* via the Newborn Screening Programme in Germany (Muller-Felber et al, 2023). Routine subsequent genetic validation analysis at the Institute of Human Genetics in Cologne, determined a compound heterozygous *SMN1* genotype with a large *SMN1* and *SMN2* deletion on one haplotype and an intragenic 4-base pair deletion in exon 7, *SMN1* ENST00000380707 c.855_858del p.(Arg288Alafs*5) and absence of any *SMN2* copy on the second haplotype (Fig. 2). The intragenic *SMN1* mutation was classified by the testing laboratory as a VUS-3A and is referred to as 855VUS in this manuscript. The 855VUS (SMN1 c.855_858del, p.Arg288Alafs*5) appears 13 times in the gnomAD database and was classified as VUS-3A based on a similar rationale as for 861VUS.

Interestingly, although independent and different, both *SMN1* variants are predicted to encode the same truncated protein, with the last seven amino acids truncated and substituted with four different amino acids before a premature stop (Fig. 2).

## Zebrafish

Zebrafish were maintained by standard protocols approved by Griffith University, approval GRIDD1122AEC. Homozygous *smn*$^{-/-}$ embryos were produced from *Tg(hs:RFP-SMN*$^{+/-}$*);smnY262stop*$^{-/-}$ zebrafish, previously published (Boon et al, 2009; Hao le et al, 2013). Incrossing this line produces 100% maternal zygotic *smnY262stop*$^{-/-}$ mutants, of which 25% do not inherit the rescue transgene *Tg(hs:RFP-SMN)*, so do not express any form of SMN protein (note that a minimal amount of maternally deposited SMN—under detection threshold—cannot be fully excluded). To select these 25% maternal zygotic animals—*smnY262stop*$^{-/-}$ (named *smn*$^{-/-}$ in this manuscript)—from their transgenic siblings—*Tg(hs:RFP-SMN*$^{+/-}$*);smnY262stop*$^{-/-}$ (named *smn*$^{-/-}$;*Tg(SMN1)* or *Tg(SMN1)* here)—the clutches were heat-shocked at 24 hpf (hours post fertilization) for 2 h at 37 °C and subsequently identified by the absence of detectable red fluorescence (Appendix Fig. S1).

## Genetic and functional analysis

Wild-type and variant *SMN1* cDNAs were synthesized as gene blocks by Gene Universal *Inc.* and inserted into a pME Gateway-compatible backbone (Appendix Table S1). Each pME-*SMN1* was further subcloned by Gateway cloning into a custom RNA expression plasmid, pT3TS-R1R3 (Addgene #140878) (Tromp et al, 2023) along with a polyadenylation sequence from the Tol2-kit plasmid 302-p3E-polyA. mRNAs were synthesized from the resultant RNA expression plasmids using a mMESSAGE mMACHINE T3 transcription kit (Invitrogen) from plasmid DNA linearized by *Kpn*I digestion. The pT3TS-R1R3 destination plasmid adds additional 5' and 3' untranslated flanking sequences to each mRNA that are designed to enhance stability (Tromp et al, 2023). mRNA was purified with a MEGAclear™ Clean-Up Kit (Invitrogen) according to the manufacturer's protocols and stored

### The paper explained

#### Problem

Spinal muscular atrophy (SMA) is a severe genetic disorder and a leading cause of infant death. While revolutionary treatments are now available, they must be administered early to prevent irreversible motor neuron loss. Newborn screening (NBS) programs help identify infants with known pathogenic mutations, enabling timely treatment. However, for novel mutations, there are no robust tools to predict the disease course before the onset of irreversible symptoms. Consequently, clinicians face a critical dilemma, whether to (i) initiate treatment pre-emptively, risking unnecessary intervention and cost, or (ii) delay and risk irreversible harm.

#### Results

We demonstrated that the zebrafish animal model can serve as a powerful diagnostic/prognostic tool to support clinical settings in assessing the pathogenicity of novel *SMN1* mutations within a time-frame that matches patients' requirements for early treatment implementation.

#### Impact

This study provides a compelling real-world example of zebrafish-based functional experiments informing clinical decision-making for human diseases, and specifically, here for SMA. Our approach directly informed patient care, avoiding unnecessary treatment in two cases and saving over US$4 million in healthcare costs. We propose that zebrafish assays be integrated to support early and accurate diagnosis.

at −80 °C. Purified mRNAs were microinjected into the yolk of 1-cell stage embryos at 250 pg final as previously described (Laird et al, 2016). Mock-injected control embryos were injected with vehicle only (0.1% phenol red in water) and did not receive any mRNA. Injections were conducted using clutches obtained from *Tg(hs:RFP-SMN*$^{+/-}$*);smnY262stop*$^{-/-}$ incrosses. Maternal zygotic mutants *smnY262stop*$^{-/-}$ (*smn*$^{-/-}$, 25%) were further separated from their sibling as described above and maintained at 28 °C to conduct the analyses (Appendix Fig. S1). Zebrafish morphology was recorded using an Olympus MVX10 microscope driven by cellSens software. Survival was recorded each morning at 9 am; death was determined by the absence of a heartbeat. To assess motor function, larvae were distributed in 24-well plates with a single animal per well and supplemented with 0.5 mL of E3 medium. Motor functions were assessed for 24 min using a ZebraBox Revolution (ViewPoint). The protocol included repetitive cycles of 4 min of light and 4 min of dark (3 cycles). The protocol was complemented with repetitive flashes of light and 1-s acoustic vibration (250 Hz) every minute (independently offset by 30 s) to assess swimming reactions/abilities.

## Quantitative real-time PCR

Total RNA was extracted from single zebrafish embryos using 100 μL of TRIzol Reagent (ThermoFisher Scientific) and a Directzol RNA MicroPrep Kit (Zymo Research), with on-column DNaseI digestion of genomic DNA, following manufacturer protocols (25 μL water elution). cDNA was synthesized from 12 μL of total RNA using a QuantiTect Reverse Transcription Kit (QIAGEN) with random hexamer primers, then diluted 1:5 with 80 μL of water. Real-time quantitative PCR was performed using a

Quantitect SYBR Green PCR kit (QIAGEN) and a RotorGene™ Q instrument (QIAGEN). 10 μL reactions contained 2 μL of diluted cDNA. Data were analyzed by relative quantitation using the comparative $C_T$ method and normalized to *dre-eef1a*. PCR primers with melting temperatures of approximately 60 °C were designed using Primer-Blast to amplify unique PCR products 100–200 bp long. Gene-specific oligonucleotide sequences are shown in Appendix Table S2.

### Western blotting

Embryos and larvae were collected and rinsed in Ringer's solution. The yolk was removed by pipetting up and down rigorously using a P200 pipette. De-yolked fish were washed with PBS and lysed in RIPA buffer (0.5 μL per fish for 1 dpf, 1 μL per fish for 3 dpf, 2 μL per fish for 5, 7, and 9 dpf). 10–25 fish were lysed per time point. Total protein concentration was measured using Pierce™ BCA Protein Assay Kits (Thermo Scientific™, Catalog number 23225). Proteins were separated on 10% SDS-PAGE gels in SDS running buffer and were transferred to a 0.45-μm hydrophobic PVDF membrane (Immobilon®-P, Merck Millipore, Catalog number: IPVH00010) for western blotting using a Bio-Rad Trans-Blot Turbo semi-dry transfer system, following the manufacturer's default protocol for 1.5 mm gels. Membranes were blocked with 5% BSA for 1 h at room temperature with shaking, then incubated with primary antibodies overnight at 4 °C with gentle shaking. Membranes were incubated with secondary antibody (1:4000 dilution) for 1 h at room temperature with gentle shaking. After initial probing, membranes were stripped twice using western blot stripping buffer (1% SDS, 1% Tween-20, 200 mM glycine, pH 2.2) (15 min each at room temperature with gentle agitation). Membranes were then washed in TBST and re-blocked in 5% BSA before re-probing. The following primary antibodies were used: anti-MANSMN12(2E6) (MANSMN12, 2E6, 1:1000; Glenn E. Morris, Developmental Studies Hybridoma Bank), and anti-β-actin (Sigma-Aldrich, Catalog number A5441-.2 ML, 1:2000); with anti-mouse IgG HRP-linked secondary antibody (Cell Signaling, 7076 s). All membranes were imaged using a ChemiDoc Imaging System (Bio-Rad), and subsequent analysis was performed using Image Lab software version 6.1.0 (Bio-Rad).

### Experimental study design and statistics

Sample sizes were determined by the number of eggs obtained per crossing on the days of the experiments, fertilization rates, and the number of maternal zygotic mutant $smn^{-/-}$ embryos identified post-heat-shock genotyping. mRNA injections were performed simultaneously by two independent researchers, each injecting all experimental conditions, and the embryos were pooled, except for VUS replicates 1 and 2, which were deliberately restricted to a single operator to assess potential inter-operator variation (Appendix Figs. S2 and S3). Measurements and analyses were performed without blinding. Statistical analysis was performed using Graph-Pad Prism 10 as described in the figure legends. Error bars represent the standard error of the mean. Exact *P* values are shown in either the main Figures or their accompanying Appendix Figures. *P* values less than 0.0001 are shown as $P < 0.0001$. All experiments were performed at least three times. ARRIVE guidelines were followed (Percie du Sert et al, 2020).

## Data availability

This study includes no data deposited in external repositories.

The source data of this paper are collected in the following database record: biostudies:S-SCDT-10_1038-S44321-025-00355-8.

## Peer review information

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

## Acknowledgements

We thank the children investigated, their families, and carers for their valuable contributions to data collection. We thank the clinical geneticist (Dr. Mert Karakaya, Univ. of Cologne) and the investigators of the genetic diagnoses (Dr. Nico Fuhrmann, Dr. Jutta Becker from Univ. of Cologne) involved in the routine diagnostic investigation of the newborns. We are grateful to Dr Danielle Kamato and Zheng Jie Chia for their assistance in establishing the Western blot assays presented in Appendix Figs. S7 and S8. We also acknowledge the funding allocated by the Australian Functional Genomics Network and the work of the Australian Functional Genomics Network Clinical and Scientific Review Committees. The Australian Functional Genomics Network is funded by the Medical Research Future Fund (Funding ID MRF2007498) and administered by the Murdoch Children's Research Institute. This study was funded by an Australian Functional Genomics Network Catalyst Grant #11501 to JG, BW & SG, a Center for Molecular Medicine Cologne Grant (C18) grant to BW, and an NHMRC Investigator fellowship No 1174145 to JG.

## Author contributions

**Brett W Stringer**: Data curation; Formal analysis; Supervision; Validation; Investigation; Visualization; Methodology; Writing—original draft; Writing—review and editing. **Yougang Zhang**: Data curation; Formal analysis; Validation; Investigation; Visualization; Methodology. **Afsaneh Taghipour-Sheshdeh**: Data curation; Formal analysis; Validation; Investigation; Visualization; Methodology. **Shuxiang Goh**: Resources; Funding acquisition; Writing—review and editing. **Heike Kölbel**: Resources; Writing—review and editing. **Michelle A Farrar**: Resources; Writing—review and editing. **Brunhilde Wirth**: Resources; Funding acquisition; Writing—review and editing. **Jean Giacomotto**: Conceptualization; Resources; Formal analysis; Supervision; Funding acquisition; Validation; Investigation; Visualization; Methodology; Writing—original draft; Project administration; Writing—review and editing.

Source data underlying figure panels in this paper may have individual authorship assigned. Where available, figure panel/source data authorship is listed in the following database record: biostudies:S-SCDT-10_1038-S44321-025-00355-8.

## Disclosure and competing interests statement

The authors declare no competing interests.

# Expanded View Figures

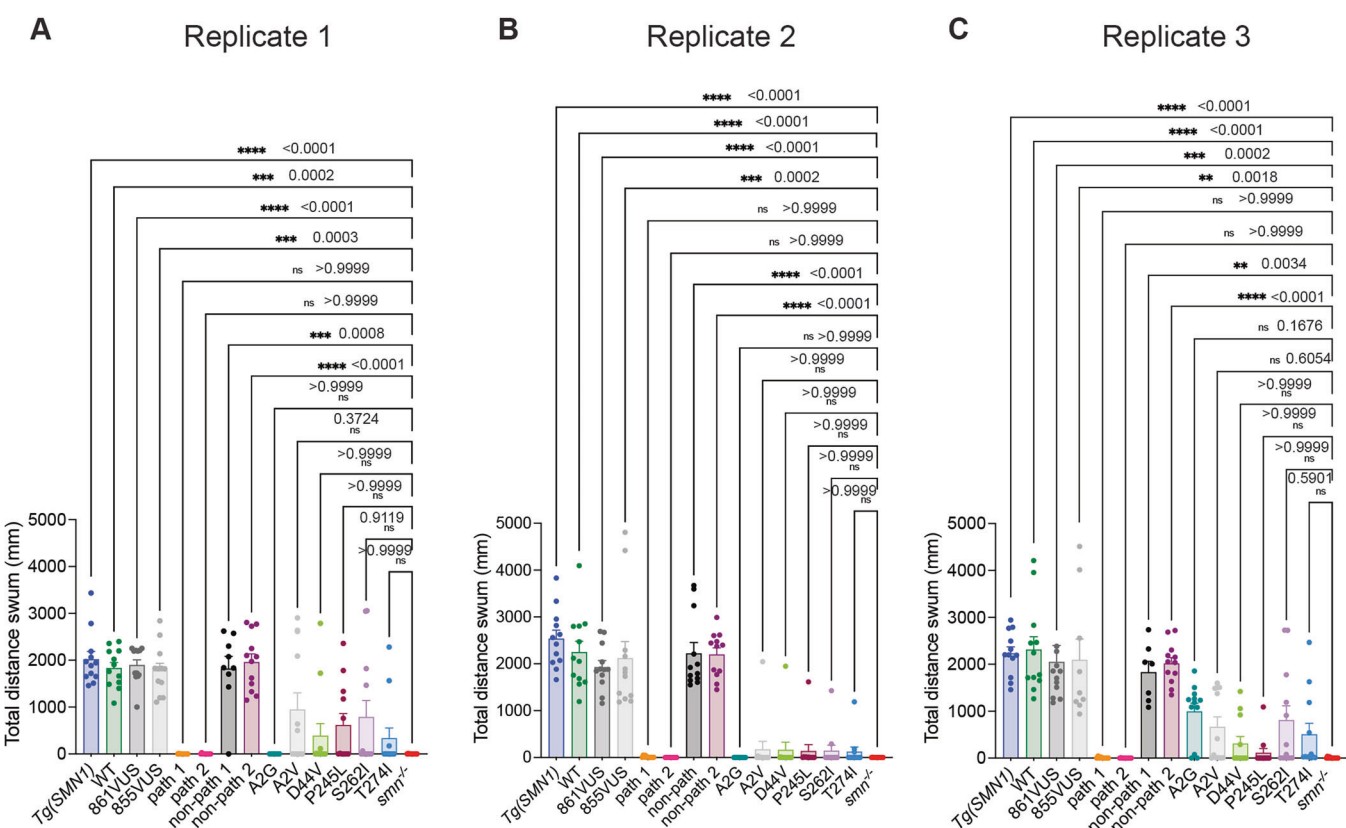

**Figure EV1.** mRNA injections of *SMN1* variants associated with SMA type II and III do not significantly improve the swimming ability of *smn*⁻/⁻ zebrafish.

(A–C) While *smn*⁻/⁻ zebrafish swam significantly better at 5 dpf when injected with *SMN1* wt mRNA (WT), non-pathogenic *SMN1* variant mRNA or mRNA from either VUS, those injected with SMA type II- or type III-associated *SMN1* variant mRNA did not. Graphs represent comparisons of the total distance swum by each cohort of fish over 24 min. Data from three replicate experiments are shown. Each data point represents one fish. Error bars represent standard error of the mean. Statistical significance was evaluated using the Kruskal–Wallis test with Dunn's correction for multiple comparisons. ****P < 0.0001; ***P < 0.001; **P < 0.01; ns, not significant. Exact P values are also shown. Source data are available online for this figure.

