## [Peer Review File · EMBO Molecular Medicine]

Clinical relevance of zebrafish for gene variants testing. Proof-of-principle with SMN1/SMA.

Brett Stringer, Yougang Zhang, Afsaneh Taghipour-Sheshdeh, Shuxiang Goh, Heike Koelbel, Michelle Farrar, Brunhilde Wirth, and Jean Giacomotto

Corresponding author(s): Jean Giacomotto (j.giacomotto@griffith.edu.au)

Review Timeline:

Submission Date:	28th Jan 25
Editorial Decision:	18th Feb 25
Revision Received:	8th Jul 25
Editorial Decision:	5th Aug 25
Revision Received:	24th Oct 25
Accepted:	24th Nov 25

Editor: Jingyi Hou

Transaction Report:

18th Feb 2025

Dear Dr. Giacomotto,

Thank you for submitting your work to EMBO Molecular Medicine. We have now heard back from the three reviewers who agreed to evaluate your manuscript. You will see from the comments below that the reviewers find the manuscript to be of interest. They raise, however, several important points, which should be convincingly addressed in a revision of this work.

The recommendations of the reviewers are rather clear, so there is no need to repeat the points listed below. In particular, all three reviewers requested more methodological details and information, and Reviewer #2 requested additional pathogenic controls, which should be addressed. Reviewer #3 is concerned that the morphological phenotype of the homozygous mutant rescued with either WT human SMN1 or non-pathogenic variant mRNA closely resembles that of the WT controls. We would ask you to respond to this concern and perform the requested experiments.

As you may already know, our editorial policy allows in principle a single round of major revision, so it is essential to provide responses to the reviewers' comments that are as complete as possible. Please feel free to contact me in case you would like to discuss in further detail any of the issues raised by the reviewers.

Please also contact us as soon as possible if similar work is published elsewhere. If other work is published, we may not be able to extend the revision period beyond three months.

I look forward to receiving your revised manuscript.

Yours sincerely,
Jingyi

Jingyi Hou
Senior Editor
EMBO Molecular Medicine

We require:

- 1) A .docx formatted version of the manuscript text (including legends for main figures, EV figures and tables). Please make sure that the changes are highlighted to be clearly visible.
- 2) Individual production quality figure files as .eps, .tif, .jpg (one file per figure). For guidance, download the 'Figure Guide PDF': (<https://www.embopress.org/page/journal/17574684/authorguide#figureformat>).
- 3) A .docx formatted letter INCLUDING the reviewers' reports and your detailed point-by-point responses to their comments. As part of the EMBO Press transparent editorial process, the point-by-point response is part of the Review Process File (RPF), which will be published alongside your paper.
- 4) A complete author checklist, which you can download from our author guidelines (<https://www.embopress.org/page/journal/17574684/authorguide#submissionofrevisions>). Please insert information in the checklist that is also reflected in the manuscript. The completed author checklist will also be part of the RPF.

6) It is mandatory to include a 'Data Availability' section after the Materials and Methods. Before submitting your revision, primary datasets produced in this study need to be deposited in an appropriate public database, and the accession numbers and database listed under 'Data Availability'. Please remember to provide a reviewer password if the datasets are not yet public (see <https://www.embopress.org/page/journal/17574684/authorguide#dataavailability>).

12) Author contributions: You will be asked to provide CRediT (Contributor Role Taxonomy) terms in the submission system. These replace a narrative author contribution section in the manuscript.

13) A Conflict of Interest statement should be provided in the main text.

14) Every published paper now includes a 'Synopsis' to further enhance discoverability. Synopses are displayed on the journal webpage and are freely accessible to all readers. They include a short stand first (maximum of 300 characters, including space) as well as 2-5 one-sentences bullet points that summarizes the paper. Please write the bullet points to summarize the key NEW findings. They should be designed to be complementary to the abstract - i.e. not repeat the same text. We encourage inclusion of key acronyms and quantitative information (maximum of 30 words / bullet point). Please use the passive voice. Please attach these in a separate file or send them by email, we will incorporate them accordingly.

Please also suggest a visual abstract to illustrate your article as a PNG file 550 px wide x 300-600 px high.

15) All Materials and Methods need to be described in the main text using our 'Structured Methods' format. According to this format, the Methods section includes a Reagents and Tools Table (listing key reagents, experimental models, software and relevant equipment and including their sources and relevant identifiers) followed by a Methods and Protocols section describing the methods, ideally using a step-by-step protocol format. The aim is to facilitate adoption of the methodologies across labs.

Please download and fill our Reagents and Tools Table template (.docx), which you can find in our author guidelines: <https://www.embopress.org/page/journal/17574684/authorguide#structuredmethods>

***** Reviewer's comments *****

Referee #1 (Comments on Novelty/Model System for Author):

The technical experiment is performed well and the results are clear there are some details that can be included for clarity. This is a simple rapid system to test at least variants of uncertain significance in SMN this is important in newborn screening and thus is novel and has high impact

Referee #1 (Remarks for Author):

The paper by Stringer et al. provides a very useful system for testing mutations that occur in SMA. The authors report 2 mutations that were detected in newborn screening, but the question was where they are pathogenic if so then treatment should be initiated. The results are very clear. However, although relatively minor there are a few points that need to be clarified in the text these are given below.

1) Can the author be absolutely clear about the mutant fish used. The smn^{-/-} fish is a mutant described by the beattie laboratory. The original paper by Boon et al reference 15 described three different zebrafish mutants. These were smn^{Y262} stop in exon 6, smn^{L264} stop and a missense mutation smn^{G264D} both in exon 7 all of these seriously disrupt function. So homozygous mutations of either stop mutation results in basically smn^{-/-} animals which live to 11-12 days on maternal SMN. The survival of these fish can be modified by using maternal zygotic mz-smn^{-/-} which are described in the Hao et al paper reference 16. With the heat shock transgene expressing human SMN but no zebrafish smn these animals lived for 9.2 days but without the transgene it was 4 days. So, the authors report the animals living for 6 days but I am not sure which mutant is used. Can the authors indicate whether this breeding uses the maternal zygotic lines or the direct mutant lines it looks like the maternal zygotic so no maternal zebrafish smn the survival is a bit longer 6 days instead of 4 but that would seem like acceptable variation. Also, for completeness it would be nice to know which specific mutant is used ie smn^{262stop}. It really does not matter unless you try to duplicate exactly the system, but it certainly would be good to have these specifics in the methods.

2) The presence of the hsp70 RFP-SMN transgene in figure 3 which gives complete rescue are these animals heat shocked that is when it rescues this should be indicated in the figure legend. This might also imply the use of mz smn^{-/-} animals in the other studies above.

3) While it again is not critical to the manuscript the authors probably should say something about the fact that all SMN missense mutations causing SMA tested to date be they mild or severe do not rescue a Smn null allele in mice or cell lines thus a true SMN missense mutation that causes SMA does not seem to have any function without some wild type SMN from SMN2 or elsewhere. See Iyer CC, et al. Mild SMN missense alleles are only functional in the presence of SMN2 in mammals. Hum Mol Genet. 2018 Oct 1;27(19):3404-3416. doi: 10.1093/hmg/ddy251. and Blatnik AJ et al. Conditional deletion of SMN in cell culture identifies functional SMN alleles. Hum Mol Genet. 2020 Nov 1;29(21):3477-3492. doi: 10.1093/hmg/ddaa229 for details. This just gives context to testing mutations that are in essence small. In theory selection in the cell line reported by Blatnik et al could

produce similar results it is just the actually fish test is very likely quicker as you do not have to select clones nor test all of them for any production of full length smn thus it will be quicker in the fish experiment as long as you have the breeding stock.

Referee #2 (Comments on Novelty/Model System for Author):

Some essential details of the system used and how results were analysed are missing, preventing appropriate assessment of technical quality.

The approach is novel, and the potential impact is very high, providing an assay for VUS assessment in SMN1 in a short time frame. Particularly important, given the treatment options available for SMA.

The main area for improvement is in the use of additional controls, with only a single severe pathogenic control used in the study. This falls short of the recommended guidelines and also does not provide confidence that mutation in ClassII-IV could be identified, raising the possibility of a false negative result and questioning how informative the zebrafish evidence was in the classification of these cases.

Referee #2 (Remarks for Author):

The manuscript describes an approach to assess VUS in SMN. Whilst the approach appears suitable there is some essential information missing from the manuscript, critical for the assessment of the approach and the interpretation of the pathogenicity of the VUS. Further methodological detail and information on the strain used is required. Most importantly only a single pathogenic control, from the most severe class of SMN1 mutations, is included. The results, whilst highly promising, therefore do not provide confidence that other classes of mutation would be identified or an understanding of how variable the results from different pathogenic mutations would be. Brnich et al., 2020 discuss the level of evidence and controls required and would be useful for the authors as a guide. Whilst 10 controls may not be required, multiple pathogenic controls across multiple classes would be a minimum.

Major concerns

1. The details of the zebrafish strain used are missing. References to other zebrafish studies are provided but it is not clear which, if any, of the referenced strains are utilised. If this is a novel strain, then proof of loss of Smn1 and characterisation of the model is necessary. If it is a previously published strain then details of the mutation should be included in the text.
2. There is no statistics section in the methods, and details of the statistical approaches used need to be included.
3. The fish are lacking pigment at later stages suggesting a background mutation or the use of a pigment inhibitor. There is no mention of either in the methods and this needs to be explained.
4. Details of the mock injections referred to in the text are not included in the methods and should be. Furthermore, the authors should ensure that all experimental methods are included in the methods section.
5. The legend refers to biological replicates, but these are also not included in the. Methods and details of how replication was conducted and how biological replicates differed needs to be included.
6. The frequency of the alleles in Gnomad should be included in the description of the cases. The Arg288Alafs*5 allele appears to be common in the database and this needs to be included and discussed.
7. The evidence that led to the variants being classified 3A initially needs to be expanded upon in the cases/patients section and details of the final classification after the inclusion of the zebrafish evidence presented in the discussion.
8. There is a single known pathogenic control used in the experiment. This appears to be a class I mutations, and this detail should be included. Critically, the use of a single pathogenic control, especially one from the most severe class of mutations is insufficient. Additional known pathogenic controls are required, including those of each mutation class. Further benign mutations would also improve the quality.
9. The text (pg 6) refers to previous data from the zebrafish SMA model, but no reference is provided, and it is not clear if this is the same model. This additional information is required.
10. The data availability statement does not meet the requirements. Sequence data and accession numbers for the plasmids need to be included. The authors are strongly encouraged to deposit all the remaining data in a repository.
11. The discussion needs to include the limitations of the study and discuss how the zebrafish evidence was used to classify the variants, alongside other clinical and population data, including the final classification of each variant.

Minor

1. Fig 2 is referred to before figure 1 and therefore the figures should be swapped in order.
2. In figure 3A & 4A there needs to be a label indicating that the numbers on the left refer to days post fertilisation.
3. The traces of the fish movements in 3C and 4C needs to be larger and of higher quality to allow their assessment.
4. The zebrafish model and its use for SMA should be introduced in the introduction.

Referee #3 (Remarks for Author):

Variant rescue in zebrafish models is not a novel approach, and I believe this article is not suitable for EMBO Molecular Medicine due to the quality of the data presented. Additionally, the morphological phenotype figures following rescue raise significant concerns. For instance, the morphology of animals rescued with WT human SMN1 and non-pathogenic variant mRNA appears nearly identical to WT controls at day 10, which seems highly improbable. Given the severity of the homozygous mutant phenotype, exogenous mRNA should not persist for that long unless they provide qPCR or Western blot. Moreover, injecting mRNA into the yolk can significantly reduce the rescue effect. The authors also fail to specify which ubiquitous promoter was used for WT-SMN1 in the Methods section. The reference they cite (Hao et al.) utilized a heatshock promoter, with the rescue effect diminishing after four days (Figure 3). Based on these concerns, I do not believe this article presents a relevant or reliable method for the research community.

Griffith Institute for Drug Discovery (GRIDD), Griffith University
Conjoint Queensland Brain Institute (QBI), The University of Queensland
Dr Jean Giacomotto, Ph.D, NHMRC Research Fellow, ZebraCLINICS Head
#Building N75 Nathan, 4111, QLD, Australia
M +61 423 071 651 @ j.giacomotto@griffith.edu.au ; j.giacomotto@uq.edu.au

21/06/2025

EMBO Molecular Medicine,

Dear Dr Jingyi Hou,

On behalf of all the authors, please find below detailed point-by-point responses to the reviewer's comments on our recent manuscript EMM-2025-21330, entitled:

“Clinical Relevance of Zebrafish for Gene Variants Testing: Proof-of-Principle with *SMNI/SMA*.”

Referee #1

Referee #1 (Comments on Novelty/Model System for Author):

The technical experiment is performed well and the results are clear there are some details that can be included for clarity. This is a simple rapid system to test at least variants of uncertain significance in SMN this is important in newborn screening and thus is novel and has high impact.

Referee #1 (Remarks for Author):

The paper by Stringer et al. provides a very useful system for testing mutations that occur in SMA. The authors report 2 mutations that were detected in newborn screening, but the question was where they are pathogenic if so then treatment should be initiated. The results are very clear. However, although relatively minor there are a few points that need to be clarified in the text these are given below.

1) Can the author be absolutely clear about the mutant fish used. The *smn*^{-/-} fish is a mutant described by the Beattie laboratory. The original paper by Boon et al reference 15 described three different zebrafish mutants. These were *smn*^{Y262} stop in exon 6, *smn*^{L264} stop and a missense mutation *smn*^{G264D} both in exon 7 all of these seriously disrupt function. So homozygous mutations of either stop mutation results in basically *smn*^{-/-} animals which live to 11-12 days on maternal SMN. The survival of these fish can be modified by using maternal zygotic *mz-smn*^{-/-} which are described in the Hao et al paper reference 16. With the heat shock transgene expressing human SMN but no zebrafish *smn* these animals lived for 9.2 days but without the transgene it was 4 days. So, the authors report the animals living for 6 days but I am not sure which mutant is used. Can the authors indicate whether this breeding uses the maternal zygotic lines or the direct mutant lines it looks like the maternal zygotic so no maternal zebrafish *smn* the survival is a bit longer 6 days instead of 4 but that would seem like acceptable variation. Also, for completeness it would be nice to know which specific mutant is used ie *smn*^{262stop}. It really does not matter unless you try to duplicate exactly the system, but it certainly would be good to have these specifics in the methods.

2) The presence of the *hsp70* RFP-SMN transgene in figure 3 which gives complete rescue are these animals heat shocked that is when it rescues this should be indicated in the figure legend. This might also imply the use of *mz smn*^{-/-} animals in the other studies above.

3) While it again is not critical to the manuscript the authors probably should say something about the fact that all SMN missense mutations causing SMA tested to date be they mild or severe do not rescue a *Smn* null allele in mice or cell lines thus a true SMN missense mutation that causes SMA does not seem to have any function without some wild type SMN from SMN2 or elsewhere. See Iyer CC, et al. Mild SMN missense alleles are only functional in the presence of SMN2 in mammals. *Hum Mol Genet.* 2018 Oct 1;27(19):3404-3416. doi: 10.1093/hmg/ddy251. and Blatnik AJ et al. Conditional deletion of SMN in cell culture identifies functional SMN alleles. *Hum Mol Genet.* 2020 Nov 1;29(21):3477-3492. doi: 10.1093/hmg/ddaa229 for details. This just gives context to testing mutations that are in essence small. In theory selection in the cell line reported by Blatnik et al could produce similar results it is just the actually fish test is very likely quicker as you do not have to select clones nor test all of them for any production of full length *smn* thus it will be quicker in the fish

experiment as long as you have the breeding stock.

Responses.

We would like to thank Reviewer #1 for his time and constructive feedback, which helped significantly improve the clarity and quality of this manuscript.

1) Can the author be absolutely clear about the mutant fish used. The *smn*^{-/-} fish is a mutant described by the Beattie laboratory. The original paper by Boon et al reference 15 described three different zebrafish mutants. These were *smn*^{Y262 stop} in exon 6, *smn*^{L264 stop} and a missense mutation *smn*^{G264D} both in exon 7 all of these seriously disrupt function. So homozygous mutations of either stop mutation results in basically *smn*^{-/-} animals which live to 11-12 days on maternal SMN. The survival of these fish can be modified by using maternal zygotic *mz-smn*^{-/-} which are described in the Hao et al paper reference 16. With the heat shock transgene expressing human SMN but no zebrafish *smn* these animals lived for 9.2 days but without the transgene it was 4 days. So, the authors report the animals living for 6 days but I am not sure which mutant is used. Can the authors indicate whether this breeding uses the maternal zygotic lines or the direct mutant lines it looks like the maternal zygotic so no maternal zebrafish *smn* the survival is a bit longer 6 days instead of 4 but that would seem like acceptable variation.

Also, for completeness it would be nice to know which specific mutant is used ie *smn*^{262stop}. It really does not matter unless you try to duplicate exactly the system, but it certainly would be good to have these specifics in the methods.

We thank the reviewer for offering us the chance to clarify the mutant and breeding methodology used. Indeed, to simplify and try to address our report to a “clinical” readership, we limited the use of complex backgrounds. We initially thought that the important part was to highlight that the experiment/test/diagnostic starts with zebrafish embryos with a complete absence of SMN protein (*smn1* total loss-of-function). The reviewer is very accurate in all his descriptions.

To address the reviewer’s comment, we have now significantly **modified our manuscript by extending the method section and by supplementing it with an Appendix Figure S1, both describing the line and breeding methodology used.**

We indeed used the mutation *smn*^{Y262stop}, first isolated in Boon *et al.* 2009 (Ref 15), that was later complemented with a rescue heat shock transgene *Tg(hs:RFP-SMN)* in Hao *et al.* 2013 (Ref 16) -also named *Tg(hsp70:RFP-SMN)* in parts of the manuscript-. The corresponding line, *Tg(hs:RFP-SMN);smn*^{Y262stop}, was used by Hao *et al.* and this study to generate maternal zygotic *smn*^{Y262stop}^{-/-} mutants, knockout for *smn1* and born without maternal SMN. As brilliantly described by reviewer #1, these maternal zygotic embryos lacking SMN protein from birth were first described in 2013 to live around 4 days-post-fertilisation (dpf). However, in our laboratory setting, these animals live up to 5/6 dpf as described in our short study, with consistent phenotypes across generations.

We have now clarified these points in the method section and provided a methodology schematic in Appendix Figure S1. We hope this addition will satisfy reviewer #1 and will help to clarify this point that seems important for a broad readership.

2) The presence of the hsp70 RFP-SMN transgene in figure 3 which gives complete rescue are these animals heat shocked that is when it rescues this should be indicated in the figure legend. This might also imply the use of *mz smn*^{-/-} animals in the other studies above.

This is indeed correct. The control name *smn*^{-/-}; *Tg(SMN1)* are embryos from *Tg(hs:RFP-SMN);smn*^{Y262stop} incrosses that have been heat shocked once at 37°C - at the same time as all other samples. This is now clarified in the legends, e.g. “Control *smn*^{-/-}; *Tg(SMN1)* were heat shocked once at 24 hpf to produce SMN” in Figure 3 and “*wt-SMN1* mRNA from an endogenous *SMN1* transgene (*Tg(SMN1)*) (activated by heat shock at 24 hpf) in

Figure 4. This is now also illustrated in Appendix Figure S1.

3) While it again is not critical to the manuscript the authors probably should say something about the fact that all SMN missense mutations causing SMA tested to date be they mild or severe do not rescue a *Smn* null allele in mice or cell lines thus a true SMN missense mutation that causes SMA does not seem to have any function without some wild type SMN from *SMN2* or elsewhere. See ¹Hum Mol Genet. 2018 Oct 1;27(19):3404-3416. doi: 10.1093/hmg/ddy251. and Blatnik AJ et al. Conditional deletion of SMN in cell culture identifies functional SMN alleles. Hum Mol Genet. 2020 Nov 1;29(21):3477-3492. doi: 10.1093/hmg/ddaa229 for details. This just gives context to testing mutations that are in essence small. In theory selection in the cell line reported by Blatnik et al could produce similar results it is just the actually fish test is very likely quicker as you do not have to select clones nor test all of them for any production of full length *smn* thus it will be quicker in the fish experiment as long as you have the breeding stock.

This is indeed a very important and interesting point. While our study was primarily designed to assess whether at least “partial SMN function” is retained - aiming to rapidly support clinical decisions regarding early treatment initiation of SMA Type I -, the potential of our approach to also identify hypomorphic variants or risk of Type II/III would represent a major added value.

To blend all Reviewer requests, we significantly extended and complemented our manuscript with the testing of additional variants associated with SMA Type II and Type III, hence, keeping aside *SMN2* copy number effect, hypothetically hypomorphs.

Interestingly, our results suggest that the assay may detect partial loss-of-function effects for these variants, which is highly encouraging. However, we remain cautious in our interpretation, especially given that, as Reviewer #1 astutely noted, existing models have so far failed to detect hypomorphic effects in the absence of residual SMN protein. Further investigation will be required to fully validate this potential. Nonetheless, we believe this initial finding lays a promising foundation for future, more in-depth studies aimed at fully exploring the diagnostic potential of our approach.

In line with Reviewer #1’s suggestion, we have now included a “Limitations and potential for other forms of SMA” section presenting these tests, their potential limitations and incorporated the comments above related to *SMN2* as well as the references mentioned; that are very useful and complementary to our study. Thank you for pointing these out.

Referee #2

Referee #2 (Comments on Novelty/Model System for Author):

Some essential details of the system used and how results were analysed are missing, preventing appropriate assessment of technical quality.

The approach is novel, and the potential impact is very high, providing an assay for VUS assessment in SMN1 in a short time frame. Particularly important, given the treatment options available for SMA.

The main area for improvement is in the use of additional controls, with only a single severe pathogenic control used in the study. This falls short of the recommended guidelines and also does not provide confidence that mutation in ClassII-IV could be identified, raising the possibility of a false negative result and questioning how informative the zebrafish evidence was in the classification of these cases.

Referee #2 (Remarks for Author):

The manuscript describes an approach to assess VUS in SMN. Whilst the approach appears suitable there is some essential information missing from the manuscript, critical for the assessment of the approach and the interpretation of the pathogenicity of the VUS. Further methodological detail and information on the strain used is required. Most importantly only a single pathogenic control, from the most severe class of SMN1 mutations, is included. The results, whilst highly promising, therefore do not provide confidence that other classes of mutation would be identified or an understanding of how variable the results from different pathogenic mutations would be. Brnich et al., 2020 discuss the level of evidence and controls required and would be useful for the authors as a guide. Whilst 10 controls may not be required, multiple pathogenic controls across multiple classes would be a minimum.

Major concerns

1. The details of the zebrafish strain used are missing. References to other zebrafish studies are provided but it is not clear which, if any, of the referenced strains are utilised. If this is a novel strain, then proof of loss of *Smn1* and characterisation of the model is necessary. If it is a previously published strain then details of the mutation should be included in the text.
2. There is no statistics section in the methods, and details of the statistical approaches used need to be included.
3. The fish are lacking pigment at later stages suggesting a background mutation or the use of a pigment inhibitor. There is no mention of either in the methods and this needs to be explained.
4. Details of the mock injections referred to in the text are not included in the methods and should be. Furthermore, the authors should ensure that all experimental methods are included in the methods section.
5. The legend refers to biological replicates, but these are also not included in the. Methods and details of how replication was conducted and how biological replicates differed needs to be included.
6. The frequency of the alleles in Gnomad should be included in the description of the cases. The Arg288Alafs*5 allele appears to be common in the database and this needs to be included and discussed.
7. The evidence that led to the variants being classified 3A initially needs to be expanded upon in the cases/patients section and details of the final classification after the inclusion of the zebrafish evidence

presented in the discussion.

8. There is a single known pathogenic control used in the experiment. This appears to be a class I mutations, and this detail should be included. Critically, the use of a single pathogenic control, especially one from the most severe class of mutations is insufficient. Additional known pathogenic controls are required, including those of each mutation class. Further benign mutations would also improve the quality.

9. The text (pg 6) refers to previous data from the zebrafish SMA model, but no reference is provided, and it is not clear if this is the same model. This additional information is required.

10. The data availability statement does not meet the requirements. Sequence data and accession numbers for the plasmids need to be included. The authors are strongly encouraged to deposit all the remaining data in a repository.

11. The discussion needs to include the limitations of the study and discuss how the zebrafish evidence was used to classify the variants, alongside other clinical and population data, including the final classification of each variant.

Minor

1. Fig 2 is referred to before figure 1 and therefore the figures should be swapped in order.

2. In figure 3A & 4A there needs to be a label indicating that the numbers on the left refer to days post fertilisation.

3. The traces of the fish movements in 3C and 4C needs to be larger and of higher quality to allow their assessment.

4. The zebrafish model and its use for SMA should be introduced in the introduction.

Responses.

We would like to thank the reviewer #2 for this in-depth review and for helping us significantly improve the quality of our manuscript.

Major concerns

1. The details of the zebrafish strain used are missing. References to other zebrafish studies are provided but it is not clear which, if any, of the referenced strains are utilised. If this is a novel strain, then proof of loss of *Smn1* and characterisation of the model is necessary. If it is a previously published strain then details of the mutation should be included in the text.

As outlined in our response to Reviewer #1 (point 1), we have substantially expanded the Methods section and included Appendix Figure S1 to provide these details, along with an illustration of the breeding/methodology strategy used to generate the maternal zygotic bi-allelic mutants analysed in this study. We hope this addition addresses the reviewer's concerns satisfactorily.

2. There is no statistics section in the methods, and details of the statistical approaches used need to be included. We have now added an *Experimental study design and statistics* section to the expanded Methods and Protocols. Details of the statistical tests used have been added to the figure legends. We hope this will address this request.

3. The fish are lacking pigment at later stages suggesting a background mutation or the use of a pigment inhibitor. There is no mentioned of either in the methods and this needs to be explained.

We did not use any pigmentation inhibitors such as anti-tyrosinase inhibitors (e.g. phenylthiourea) or any other agents. These animals do develop full pigmentation as they age (horizontal stripes). Following this comment, we looked back at our pictures and past projects. We agree that their lateral sides do show some apparent lack of pigmentation when compared to some other lines we use. This is not something that we took notice of before.

While we don't have a definite explanation for this, we believe it may simply be due to their genetic background that has involved mixing their original genetic AB background with TL long-fins background from one of our Tg(MN:GFP) marker line (which also present an apparent delayed pigmentation when observed on their lateral side). All *Tg(hs:RFP-SMN);smnY262stop* lines used in this study exhibit long fins with horizontal stripes. Therefore, the variation in lateral pigmentation may simply reflect this particular genetic combination.

4. Details of the mock injections referred to in the text are not included in the methods and should be. Furthermore, the authors should ensure that all experimental methods are included in the methods section. Mock injections were performed as described in the cited reference and did not contain mRNA. We have now added this detail to the Methods and Protocols section. All experimental methods were also revised and significantly expanded. Thank you for bringing this to our attention.

5. The legend refers to biological replicates, but these are also not included in the. Methods and details of how replication was conducted and how biological replicates differed needs to be included. Thank you for pointing this out. We have now clarified what was meant by biological replicates in the legend of Figures 3 and 4 and added extra information concerning the number of times experiments were performed in the Methods and Protocols section.

6. The frequency of the alleles in Gnomad should be included in the description of the cases. The Arg288Alafs*5 allele appears to be common in the database and this needs to be included and discussed. To match the reviewer's request, we have now extended the presentation of each case in the method section. *Ex for the first case (861VUS):*

*"This 861VUS (SMN1c.861_864del (p.Arg288Alafs*5)) has been reported 47 times in the gnomAD database from 807,162 exomes and genomes (gnomADTM, June 2025). The VUS3A classification was primarily based on its low frequency in gnomAD controls, and the fact that it is not located within a functional domain, mutational hotspot, or highly conserved region, nor in a region typically associated with nonsense-mediated decay."*

We have also supplemented the beginning of the results section with the following statement/information: *"Both SMN1-VUS were found in heterozygous but not homozygous state with an allele frequency of 0.000029 and 0.000008, respectively in the Genome Aggregation Database (gnomADTM) v4 database, including exomes and genomes from 807,162 control individuals; functional prediction programs ranked them as likely pathogenic (<https://gnomad.broadinstitute.org/>)."*

We hope this is satisfactory.

7. The evidence that led to the variants being classified 3A initially needs to be expanded upon in the cases/patients section and details of the final classification after the inclusion of the zebrafish evidence presented in the discussion.

Classification as 3A – As detailed in point 6 above, we have now expanded the methods section for each case. This study was not intended to (re)classify these alleles per se, but rather to provide supporting evidence and tools for clinicians involved in diagnosing severe forms of SMA, particularly when rapid decisions about treatment implementation are needed. In our cases, based on zebrafish functional assays (including comparisons to known pathogenic variants) and available clinical data at the time, we concluded that the variants were non-pathogenic and that the patients were not at risk of developing early-onset SMA. Accordingly, the decision was made not to initiate treatment. However, the decision was also made to delay formal reclassification of these variants in order to accumulate more clinical evidence regarding their potential link to later-onset or milder forms—although this appears unlikely given current clinical accessibility and upcoming data. A follow-up study incorporating long-term clinical follow-up and robust epidemiological analysis (beyond the scope of the present manuscript due to word and guideline limits) is planned for release

in 2026. At that time, clinicians can contact the associated laboratories for formal reclassification of the variants as benign.

Given the relevance of this approach for clinicians globally and its alignment with widely implemented NBS programs, we believe early publication of this study is warranted. Our data, tools, and assays are already being discussed among clinicians at prestigious conferences, including:

- i) the European Human Genetics Conference (ESHG, May 2025);
- ii) the 2025 Annual SMA Conference (June 2025); and
- iii) the 30th Annual International Congress of the World Muscle Society, held at the Austria Center in Vienna.

8. There is a single known pathogenic control used in the experiment. This appears to be a class I mutations, and this detail should be included. Critically, the use of a single pathogenic control, especially one from the most severe class of mutations is insufficient. Additional known pathogenic controls are required, including those of each mutation class. Further benign mutations would also improve the quality.

We would like to thank the reviewer for the opportunity to expand our work and improve the overall quality of the manuscript. We have now incorporated seven additional SMN1 variants (six pathogenic, one non-pathogenic), including two associated with SMA Type II, four with Type III, and one known non-pathogenic variant. These are now summarised in Appendix Figure EV1, along with the supporting literature. In addition, the manuscript has been expanded to include a dedicated paragraph titled “Limitations and potential for other forms of SMA.”

Just a note regarding the selection of these variants. While several *SMN1*-variants have been associated with SMA Type II or III in the literature, we believe that caution should be taken to formally classify them as Type II or III *per se*. Indeed, the genetic background of the disease (involving highly variable *SMN2* copy numbers) would impact the disease course from patient to patient in the presence of the same allele (rendering a definite classification of these variants difficult and potentially misleading). To our knowledge, no hypomorphic variant has ever been identified in the complete absence of *SMN2* (*i.e.* all reported cases involve one or several *SMN2* copy). To try addressing the reviewer’s request and at the same time generate relevant conclusions, we reviewed the literature to carefully select variants specifically associated with only 1x *SMN2* copy, thereby enabling a tentative comparative interpretation of the outcomes.

While our manuscript and the methodology described are primarily intended for a clinical audience (aiming to highlight the utility of the zebrafish assay in urgent SMA Type I diagnosis), it has become apparent that the same approach may also have potential in detecting hypomorphic variants. That said, we have opted to present these findings with caution, in a clearly delineated section associated with the study’s limitations. This is to avoid confusion with the primary goal of the report and to ensure that readers do not mistakenly assume the method is already validated for use in characterising hypomorphs. Additional investigation will be required before such use can be formally established.

9. The text (pg 6) refers to previous data from the zebrafish SMA model, but no reference is provided, and it is not clear if this is the same model. This additional information is required.

At this stage, our laboratory uses 3 different models of SMA that have been previously published by us or by Prof Christine Beattie’s group, all presenting low to complete absence of SMN protein and presenting with comparable phenotype.

The sentence “*As previously shown, and replicated here, absence of Smn/SMN protein in the smn1^{-/-} SMA zebrafish model triggered morphological malformations from 3 days post fertilization (dpf), with larvae being smaller in size and presenting body curvature and small eyes*” was however referring to *Tg(hs:RFP-SMN);smnY262stop*. published by Hao *et al.* (Ref 16).

We have now added the associated reference 16 at the end of this sentence.

10. The data availability statement does not meet the requirements. Sequence data and accession numbers for the plasmids need to be included. The authors are strongly encouraged to deposit all the remaining data in a repository.

The Data availability statement has been revised to: “Data presented in this study are available in the accompanying Source Data”.

11. The discussion needs to include the limitations of the study and discuss how the zebrafish evidence was used to classify the variants, alongside other clinical and population data, including the final classification of each variant.

We have now included a limitations section and incorporated the testing of hypomorphs as well.

As briefly noted above in point 7, we wish to exercise caution regarding formal classification. At this stage, we have not formally reclassified these variants. The primary objective of this manuscript is to provide a rapid functional tool/assay to support clinicians in diagnosing the severe forms of SMA; specifically, to aid decisions about whether to initiate treatment or safely delay it. In the two cases presented here, treatment was not initiated, however, continued clinical monitoring is recommended in case later-onset symptoms emerge. A follow-up study incorporating clinical and epidemiological data is expected in 2026, at which point reclassification as benign may be considered and discussed with the relevant diagnostic laboratories.

Importantly, the extension of our study to include hypomorphs (Appendix Fig. EV1) has yielded highly encouraging results (thanks to Reviewer #2 for this insightful suggestion). This is particularly relevant given the widespread and growing implementation of newborn screening programs. However, we again wish to emphasise that our intention is not to use this assay for formal variant classification at this stage, but rather as a support tool for early clinical decision-making.

Minor

1. Fig 2 is referred to before figure 1 and therefore the figures should be swapped in order.

All figures are now cited in order.

2. In figure 3A & 4A there needs to be a label indicating that the numbers on the left refer to days post fertilisation.

“dpf” has been supplemented on top of the days/numbers to match this request.

3. The traces of the fish movements in 3C and 4C needs to be larger and of higher quality to allow their assessment.

We have now updated both figures 3 and 4 to introduce larger high-quality snapshots. We hope this will be satisfactory.

4. The zebrafish model and its use for SMA should be introduced in the introduction.

To match Reviewer #2 request, keeping in mind the manuscript length restriction, we have now complemented the introduction with the following paragraph (Changes in *italics*).

Clinicians and patients would strongly benefit from the development of innovative methodology to help support VUS resolution. *The zebrafish animal model holds tremendous promise in supporting clinical efforts to functionally characterise human disease-associated mutations¹³. Owing to its conserved neural circuitry, rapid development, and early spontaneous motor activity, zebrafish is particularly well suited for investigating neurological and neuromuscular disorders¹⁴. Zebrafish have been extensively used to study SMA, and a variety of models exist -from maternal zygotic mutants fully deprived from SMN function (used in this study)^{15,16} to partial-LOF and tissue-specific models¹⁷⁻¹⁹-. SMA in zebrafish triggers developmental*

regression, rapid motor decline, and early lethality, all taking place in the first week of life, making them an ideal model for rapid in vivo functional testing of SMNI variants. Here, we demonstrate that functional/complementation assays using zebrafish are a powerful complementary readout to support clinical decisions for the early-onset SMA forms, and, most importantly, fit within a timeframe that is congruent with pathophysiology (Fig. 1).

Referee #3

Referee #3 (Remarks for Author):

Variant rescue in zebrafish models is not a novel approach, and I believe this article is not suitable for EMBO Molecular Medicine due to the quality of the data presented. Additionally, the morphological phenotype figures following rescue raise significant concerns. For instance, the morphology of animals rescued with WT human SMN1 and non-pathogenic variant mRNA appears nearly identical to WT controls at day 10, which seems highly improbable. Given the severity of the homozygous mutant phenotype, exogenous mRNA should not persist for that long unless they provide qPCR or Western blot. Moreover, injecting mRNA into the yolk can significantly reduce the rescue effect. The authors also fail to specify which ubiquitous promoter was used for WT-SMN1 in the Methods section. The reference they cite (Hao et al.) utilized a heatshock promoter, with the rescue effect diminishing after four days (Figure 3). Based on these concerns, I do not believe this article presents a relevant or reliable method for the research community.

Responses.

We would like to thank Reviewer #3 for giving us the opportunity to clarify the relevance of our work.

1. Variant rescue in zebrafish models is not a novel approach.

We would like to thank the reviewer for taking the time to review our report. We fully agree that variant rescue in zebrafish is not a novel methodology, and our study was not intended to claim otherwise. Rather, we aimed to provide a demonstration of real-life translation/impact and clinical relevance for variant testing to increase the credibility and deployment of zebrafish research in medicine and clinical settings.

To the best of our knowledge, while mRNA/variant injection is widely used in the field, its application to directly support clinical diagnosis and treatment decisions, particularly within a timeframe compatible with patient care when no other robust tool exists, has not been previously demonstrated, and there is a clear need for robust supporting literature in this area.

Here, we present a functional tool to guide diagnosis and treatment decisions for SMA Type I, where no robust alternatives currently exist and timing is critical for patient outcomes. We believe this study provides a compelling proof-of-concept for integrating zebrafish assays into clinical workflows, especially in the context of newborn screening and VUS resolution. Beyond SMA, we hope this demonstration will contribute to increasing the credibility and clinical adoption of zebrafish models, which -to our knowledge- remain underutilised or perceived as lacking translational value when compared to their rodent counterparts.

2. The morphological phenotype figures following rescue raise significant concerns. For instance, the morphology of animals rescued with WT human SMN1 and non-pathogenic variant mRNA appears nearly identical to WT controls at day 10, which seems highly improbable. Given the severity of the homozygous mutant phenotype, exogenous mRNA should not persist for that long unless they provide qPCR or Western blot.

We thank the reviewer for this important comment. We agree that observing some animals with a morphology nearly indistinguishable from wild-type or transgenic controls at 10 dpf is outstanding. However, the results are both clear and highly reproducible across multiple experiments. That said, such a degree of “rescue” is consistent with the literature. As shown by Hao *et al.* 2013, zygotic *smn*^{-/-} mutants (from heterozygote *smn*^{+/-}

incrosses) can survive up to 11-13 dpf due to maternal deposition of zebrafish *smn1* mRNA (Ref #16, figure 3 or caption below). This is in contrast to maternal zygotic “*mz-smn^{-/-}*” mutants, which die much earlier (~4 dpf in Hao *et al.* 2013, and ~5–6 dpf in our laboratory settings). Again, this major survival extension is simply due to the inheritance/deposition of maternal mRNA within the embryos (supplementation/deposition that is in principle, reproduced by our human *SMN1* mRNA injections).

Similar results have also been published by Boon *et al.* 2009, Fig. 2 and Fig. 3 (Ref #15) -with Fig. 2 demonstrating no significant morphological changes at 11 dpf except a slightly smaller size and Fig. 3 evidencing that maternal deposition in these zygotic bi-allelic *smn^{-/-}* mutants led to detectable SMN protein up to 8dpf-.

All in all, our human *SMN1* mRNA injections into maternal zygotic mutants “*mz-smn^{-/-}*” embryos certainly recapitulate/mimic this maternal zebrafish *smn1* mRNA deposition; thus, such an outcome is consistent with prior evidence and not entirely unexpected.

Caption from Hao et al. 2013 (Ref #16)

Figure 3. Characterization of *mz-smn^{-/-}* fish. *mz-smn^{-/-}* embryos and larvae either with the *Tg(hsp70:RFP-SMN)* transgene (tg) or without. (A) Lateral view of embryos/larvae at 2, 4, 7 and 10 dpf. (B) Survival of the zygotic *smn^{-/-}* fish (red line, $n = 11$, mean = 11 ± 2.2), *mz-smn^{-/-}* + tg fish (green line, $n = 9$, mean survival = 9.6 ± 1.7) and *mz-smn^{-/-}* without the transgene (blue line, $n = 40$, mean survival = 4). (C) Levels of RFP-SMN independent of heat shock in *mz-smn^{-/-}* + tg fish and (D) *mz-smn^{-/-}* fish. Top blot in (D) is an overexposure of the RFP-SMN lane. Data are mean \pm SD.

To address the reviewer’s concern directly, we also have conducted RT-qPCR and Western blot analyses, now included in **Appendix Figure S2**. Exogenous mRNA was detectable up to 9 dpf (last timepoint analysed), with a sharp decline after 1 dpf. While mRNA integrity may be compromised at later timepoints, the Western blots clearly demonstrates the presence of residual full-length human SMN protein at 3, 5, 7, and 9 dpf, albeit at low levels past 5 dpf. The variability in protein levels across samples is likely due to inter-individual differences associated with the injections.

We hope that these clarifications, supporting literature, and additional experimental data will adequately address the reviewer’s concern.

3. Moreover, injecting mRNA into the yolk can significantly reduce the rescue effect.

Thank you for this recommendation. We have historically performed yolk injections for mRNA, and cell injections for Tol2–plasmid complexed transposase (for transgenesis). We acknowledge the potential advantage of one-cell stage cytoplasmic mRNA injection and will explore this approach in future experiments to assess whether it improves rescue efficiency. For consistency and reproducibility, all our experiments have been conducted with yolk injections.

4. The authors also fail to specify which ubiquitous promoter was used for WT-SMN1 in the Methods section. The reference they cite (Hao et al.) utilized a heatshock promoter, with the rescue effect diminishing after four

days (Figure 3).

We indeed used the *Tg(hs:RFP-SMN);smnY262stop* presented in Hao et al. 2013 (Ref#16), which incorporates a heat shock *hsp70* inducible promoter. As explained in detail in Reviewer #1 point 1, the method section has now been extensively extended and complemented with an Appendix Figure 01. The text/figures now present this information along with comprehensive details about the lines and breeding/methodology used. We hope this will be satisfactory.

Yours sincerely,

Jean Giacomotto

5th Aug 2025

Dear Dr. Giacomotto,

Thank you for submitting the revised version of your manuscript. We have now received feedback from the two reviewers who re-evaluated your study. As you will see below, Reviewer #1 is satisfied with the revisions you have made. However, Reviewer #2 has raised several remaining concerns.

In particular, Reviewer #2 notes issues related to the number of replicates and the absence of additional pathogenic controls, especially for class II-IV variants. We kindly ask that you respond to these points and include a more explicit discussion of the utility and potential limitations of the proposed assay, particularly regarding its suitability for variant classification. All remaining technical issues must be carefully addressed, including controls, statistics, detailed information on blinding and randomization, missing data points and data exclusion.

On a more editorial level:

1. Remove all figures from the manuscript file.
2. Move "The paper explained" section into the manuscript file.
3. Please ensure that all figure panels are called out in the text and in a sequential order.
4. Please resolve the discrepancy in the author's name: Afsaneh Taghipour-Sheshdeh appears in the manuscript text, while Afsaneh Taghipoursheshdeh is listed in the submission system.
5. Please merge the Funding section with the Acknowledgements section.
6. Reuse of the same cell image in Figure 1B and Figure 4A must be explicitly noted in the figure legend.
7. References: citations need to be in a alphabetical order. List 10 co-authors of a paper before adding et al. in the reference list.
8. Data availability: since this study does not generate large-scale datasets, please only include the following sentence in this section- "This study includes no data deposited in external repositories".
9. "Competing interests" should be renamed to "Disclosure statement and competing interests".
10. Please address the following issues with figure legends:
 - Please note that the exact p values are not provided in the legends of figures 3B, D; 4B, D; EV1 C
 - Please note that the error bars are not defined in the legends of figures 3B, 4B, S2 A
11. Please correct the order and headings of the manuscript sections to the following : Abstract, Introduction, Results and Discussion, Methods, Acknowledgements, Disclosure and competing interests statement, References, Figure legends, Expanded View Figure legends
12. I have modified the synopsis text (see attached). Please let me know if this is fine as is or if you would like to introduce further modifications.

We look forward to seeing a revised form of your manuscript as soon as possible.

Kind regards,
Jingyi

Jingyi Hou
Senior Editor
EMBO Molecular Medicine

***** Reviewer's comments *****

Referee #1 (Comments on Novelty/Model System for Author):

The authors completely answer all the questions asked in review and the system can clearly rapidly resolve whether a SMN mutation disrupts function of SMN or not thus allowing the determination of what is a disease causing allele or just a harmless variant.

Referee #1 (Remarks for Author):

The authors have addressed all critiques and all questions. The paper is significant.

Referee #2 (Comments on Novelty/Model System for Author):

The experiments are completed in duplicate only, preventing the use of replicate in the analysis, and the level of replication not indicated.

For the experiments in EV1 the data is compared to the WTmRNA injected animals in a different set of experiments and should be compared to data in the same experiment.

The report describes the use of zebrafish to investigate two VUS, but the classification of the VUS was not impacted by the experiments conducted. Therefore, whilst the approach shows promise to aid in variant analysis, it does not have the same impact as if it had informed classification.

Referee #2 (Remarks for Author):

Many important details in the methodology have now been included but there are multiple areas of concern still.

In the data provided two data points were excluded in 3B from the smn^{-/-} as 'spurious recordings'. Information on why these were excluded needs to be provided. Further, there is no indication that researchers were blind to the genotype when analysing or excluding data and discussion of blinding, or a statement that it did not occur needs to be included.

The injections were conducted in a randomised sequence - how were they randomised?

Two researchers conducted injections - did they each conduct an independent replicate of each sample? Researcher or replicate has not been considered in the analysis.

Although the experiment was conducted twice there is no indication as to which data came from which replicate and the lack of a third replicate prevents the consideration of replicate as a factor. In what way were replicates different, what is the level of replication? A third replicate should be completed and power analysis used to determine n-numbers (eg only 6 control fish in fig4B, may not be sufficient given variation).

In fig 4 wt mRNA injected animals were present in the survival analysis? Why were these not analysed for swimming?

Also in figure 4 there are 12 Tg(SMN1) fish at day 6 and 7 but only 6 at day 5 - why is the data from the other fish missing or excluded?

The error bars in Figure 4B are not aligned with the graphs or are missing. It is not indicated in any of the figure or legends what the error bars represent.

The additional data in EV1 does provide some encouragement that the conclusion that the variants analysed may not be pathogenic may have been correct.

Additional pathogenic controls were requested for the experiments, including those of each mutation class. No additional type 1 pathogenic controls were included, and the additional variants of the other classes were analysed independently of the VUS.

Whilst the results suggest that the assay may be able to detect reduced function in these variants compared to wildtype, it prevented comparison of the two VUS to class II-IV variants. Analysis of swimming was not conducted as for the other variants. This should be included. Furthermore, it appears that the new data was compared to WT control data from a different experiment? The controls should be from the same experiment and therefore this would need to be repeated. Given the degradation of injected RNA over time caution should be used in the interpretation of the survival data at these later timepoints, again increasing the value of the earlier swimming analyses.

In the author's response they state that details of the mock-injections are provided in the reference, but no reference is provided.

The new data is provided in the discussion section and should be incorporated into the results section.

In summary there has been some increase in the level of detail necessary and the requested experiments were partially completed, however, there are still significant concerns over lack of sufficient replication, data exclusion, absence, and analysis. Furthermore, the contribution of the zebrafish evidence to the decision not to treat is not clear, with it having no effect on variant classification.

Griffith Institute for Drug Discovery (GRIDD), Griffith University
Conjoint Queensland Brain Institute (QBI), The University of Queensland
Dr Jean Giacomotto, Ph.D, NHMRC Research Fellow, ZebraCLINICS Head
#Building N75 Nathan, 4111, QLD, Australia

25/10/2025

EMBO Molecular Medicine,**Dear Dr Jingyi Hou,**

On behalf of all the authors, please find below detailed point-by-point responses to the reviewer's comments on our recent manuscript EMM-2025-21330, entitled:

“Clinical Relevance of Zebrafish for Gene Variants Testing: Proof-of-Principle with *SMNI/SMA*.”

Major updates:

-Triplicates: We conducted three new independent experimental replicates corresponding to the previous Figures 3 and 4, provided the associated datasets, and updated the figures accordingly. Power analyses were performed to determine the minimum sample size required, and the manuscript was revised to include details on replication and statistical power. To facilitate direct comparison between both VUS, the data have been consolidated into a single, comprehensive “Figure 3.” All raw datasets are now available, with individual replicates presented separately in the supplemental material (**Appendix Figures S2–S4**).

-Hypomorphs: We have conducted new experiments corresponding to the previous Figure EV1, now including all relevant controls, VUS, and additional variants, as well as a new pathogenic Type I variant as requested. All experiments were performed in triplicate and now include motor function analyses as well. Given the scope and volume of these new data, the results have been reorganized into three main figures (Figures 4 and 5, and EV1) and two supplemental figures (Appendix Figures S5–S6). To improve clarity and align with the reviewer's recommendation, this section has also been relocated within the Results under a specific sub-heading.

-Limitation and Variant reclassification: We have now extended the manuscript to include a limitation section after the discussion and further discuss treatment decision and formal reclassification.

-Editor and Reviewer Requests: We have updated the manuscript as per the Editor and Reviewer's requests (Detailed below).

Kind regards,

Yours sincerely,

Jean Giacomotto

Editor's Requests.

Dear Dr. Giacomotto,

Thank you for submitting the revised version of your manuscript. We have now received feedback from the two reviewers who re-evaluated your study. As you will see below, Reviewer #1 is satisfied with the revisions you have made. However, Reviewer #2 has raised several remaining concerns.

In particular, Reviewer #2 notes issues related to the number of replicates and the absence of additional pathogenic controls, especially for class II-IV variants. We kindly ask that you respond to these points and include a more explicit discussion of the utility and potential limitations of the proposed assay, particularly regarding its suitability for variant classification. All remaining technical issues must be carefully addressed, including controls, statistics, detailed information on blinding and randomization, missing data points and data exclusion.

On a more editorial level:

1. Remove all figures from the manuscript file.
2. Move "The paper explained" section into the manuscript file.
3. Please ensure that all figure panels are called out in the text and in a sequential order.
4. Please resolve the discrepancy in the author's name: Afsaneh Taghipour-Sheshdeh appears in the manuscript text, while Afsaneh Taghipoursheshdeh is listed in the submission system.
5. Please merge the Funding section with the Acknowledgements section.
6. Reuse of the same cell image in Figure 1B and Figure 4A must be explicitly noted in the figure legend.

7. References: citations need to be in a alphabetical order. List 10 co-authors of a paper before adding et al. in the reference list.

8. Data availability: since this study does not generate large-scale datasets, please only include the following sentence in this section- "This study includes no data deposited in external repositories".

9. "Competing interests" should be renamed to "Disclosure statement and competing interests".

10. Please address the following issues with figure legends:

- Please note that the exact p values are not provided in the legends of figures 3B, D; 4B, D; EV1 C

- Please note that the error bars are not defined in the legends of figures 3B, 4B, S2 A

11. Please correct the order and headings of the manuscript sections to the following : Abstract, Introduction, Results and Discussion, Methods, Acknowledgements, Disclosure and competing interests statement, References, Figure legends, Expanded View Figure legends

12. I have modified the synopsis text (see attached). Please let me know if this is fine as is or if you would like to introduce further modifications.

We look forward to seeing a revised form of your manuscript as soon as possible.

Kind regards,
Jingyi

Jingyi Hou
Senior Editor
EMBO Molecular Medicine

Responses to Editor's Requests.

1. Remove all figures from the manuscript file.

Figures have now been removed.

2. Move "The paper explained" section into the manuscript file.

“The paper explained” is now presented on the third page of the manuscript.

3. Please ensure that all figure panels are called out in the text and in a sequential order.

All figure panels now called out in the text and in sequential order.

4. Please resolve the discrepancy in the author's name: Afsaneh Taghipour-Sheshdeh appears in the manuscript text, while Afsaneh Taghipoursheshdeh is listed in the submission system.

We believe this is now fixed.

5. Please merge the Funding section with the Acknowledgements section.

These two sections have now been merged.

6. Reuse of the same cell image in Figure 1B and Figure 4A must be explicitly noted in the figure legend.

The datasets and images now presented in Figure 3 (previously Figures 3 and 4) are from entirely new, independent triplicate experiments. Consequently, the images shown in Figure 1B are now unique and not reused elsewhere in the manuscript. We believe that the previous reuse of the same cell image in Figures 1B and 4A is therefore no longer applicable.

7. References: citations need to be in a alphabetical order. List 10 co-authors of a paper before adding et al. in the reference list.

The reference list has now been fully revised and updated.

8. Data availability: since this study does not generate large-scale datasets, please only include the following sentence in this section- "This study includes no data deposited in external repositories".

This section has now been updated as per the Editor's request.

9. "Competing interests" should be renamed to "Disclosure statement and competing interests".

This section has now been renamed. Note that we used “Disclosure and competing interests statement” instead of "Disclosure statement and competing interests".

10. Please address the following issues with figure legends:

- Please note that the exact p values are not provided in the legends of figures 3B, D; 4B, D; EV1 C

- Please note that the error bars are not defined in the legends of figures 3B, 4B, S2 A

Exact *P*-values are now provided for all figures, either in the main Figure or the accompanying Appendix Figure. *P*-values less than 0.0001 are shown $P < 0.0001$. Error bars have now been defined in all legends.

11. Please correct the order and headings of the manuscript sections to the following : Abstract, Introduction, Results and Discussion, Methods, Acknowledgements, Disclosure and competing interests statement, References, Figure legends, Expanded View Figure legends

The order and heading of the manuscript have now been updated. The previous Sub-heading “Discussion” has been changed to “Interpretation of findings” as now part of “Results and Discussion”.

12. I have modified the synopsis text (see attached). Please let me know if this is fine as is or if you would like to introduce further modifications.

This is fine. Thank you very much for your review and input.

Referee #1 Comments

Referee #1 (Comments on Novelty/Model System for Author):

The authors completely answer all the questions asked in review and the system can clearly rapidly resolve whether a SMN mutation disrupts function of SMN or not thus allowing the determination of what is a disease causing allele or just a harmless variant.

Referee #1 (Remarks for Author):

The authors have addressed all critiques and all questions. The paper is significant.

Responses to Reviewer #1's Comments.

We thank the reviewer for their positive feedback and for acknowledging the significance of our work.

Referee #2 Comments

Referee #2 (Comments on Novelty/Model System for Author):

The experiments are completed in duplicate only, preventing the use of replicate in the analysis, and the level of replication not indicated.

For the experiments in EV1 the data is compared to the WTmRNA injected animals in a different set of experiments and should be compared to data in the same experiment.

The report describes the use of zebrafish to investigate two VUS, but the classification of the VUS was not impacted by the experiments conducted. Therefore, whilst the approach shows promise to aid in variant analysis, it does not have the same impact as if it had informed classification.

Referee #2 (Remarks for Author):

Many important details in the methodology have now been included but there are multiple areas of concern still.

In the data provided two data points were excluded in 3B from the smn^{-/-} as 'spurious recordings'.

Information on why these were excluded needs to be provided. Further, there is no indication that researchers were blind to the genotype when analysing or excluding data and discussion of blinding, or a statement that it did not occur needs to be included.

The injections were conducted in a randomised sequence - how were they randomised?

Two researchers conducted injections - did they each conduct an independent replicate of each sample? Researcher or replicate has not been considered in the analysis.

Although the experiment was conducted twice there is no indication as to which data came from which replicate and the lack of a third replicate prevents the consideration of replicate as a factor. In what way were replicates different, what is the level of replication? A third replicate should be completed and power analysis used to determine n-numbers (eg only 6 control fish in fig4B, may not be sufficient given variation).

In fig 4 wt mRNA injected animals were present in the survival analysis? Why were these not analysed for swimming?

Also in figure 4 there are 12 Tg(SMN1) fish at day 6 and 7 but only 6 at day 5 - why is the data from the

other fish missing or excluded?

The error bars in Figure 4B are not aligned with the graphs or are missing. It is not indicated in any of the figure or legends what the error bars represent.

The additional data in EV1 does provide some encouragement that the conclusion that the variants analysed may not be pathogenic may have been correct.

Additional pathogenic controls were requested for the experiments, including those of each mutation class. No additional type 1 pathogenic controls were included, and the additional variants of the other classes were analysed independently of the VUS. Whilst the results suggest that the assay may be able to detect reduced function in these variants compared to wildtype, it prevented comparison of the two VUS to class II-IV variants. Analysis of swimming was not conducted as for the other variants. This should be included. Furthermore, it appears that the new data was compared to WT control data from a different experiment? The controls should be from the same experiment and therefore this would need to be repeated. Given the degradation of injected RNA over time caution should be used in the interpretation of the survival data at these later timepoints, again increasing the value of the earlier swimming analyses.

In the author's response they state that details of the mock-injections are provided in the reference, but no reference is provided.

The new data is provided in the discussion section and should be incorporated into the results section.

In summary there has been some increase in the level of detail necessary and the requested experiments were partially completed, however, there are still significant concerns over lack of sufficient replication, data exclusion, absence, and analysis. Furthermore, the contribution of the zebrafish evidence to the decision not to treat is not clear, with it having no effect on variant classification.

Responses to Reviewer #1's Comments.

Referee #2 (Comments on Novelty/Model System for Author):

The experiments are completed in duplicate only, preventing the use of replicate in the analysis, and the level of replication not indicated.

To fully address the reviewer's request, we have performed an entirely new series of experiments for both the VUS and the hypomorphic variants, each conducted in triplicate.

For the VUS, the previously independent assays conducted for 861VUS and 855VUS were now performed within the same experimental batches (in triplicate) to facilitate direct comparison. These datasets have been combined into a single Figure 3 (previously Figures 3 and 4 for 861VUS and 855VUS, respectively), and the individual replicate experiments are provided in Appendix Figures S2–S4, with corresponding source data available online. Power calculations, replication and minimum sample size determinations are present in the main text, page 7, and briefly detailed below.

For the experiments in EV1 the data is compared to the WTmRNA injected animals in a different set of experiments and should be compared to data in the same experiment.

To address the reviewer's concern and better implement the recommendations outlined by Brnich *et al.* (2019) -cited in page 8-, we have now fully redone and substantially expanded the previous Figure EV1 (hypomorphic variants introduction/comparison). All conditions, including controls, variants, and requested

new additional type I pathogenic control, were assessed within the same experimental batches, conducted in triplicate and supplemented with motor function analyses.

Notably, the inclusion of the two VUS together with the hypomorphic variants resulted in two complete sets of triplicate experiments for the VUS, all yielding consistent outcomes.

Given the scope and amount of data, the new results associated with the hypomorphic variants (previously in Figure EV1) have been reorganized into three main figures (Figures 4 and 5, and EV1) and two supplemental figures (Appendix Figures S5–S6).

The report describes the use of zebrafish to investigate two VUS, but the classification of the VUS was not impacted by the experiments conducted. Therefore, whilst the approach shows promise to aid in variant analysis, it does not have the same impact as if it had informed classification.

Our initial data, plus now the extension to hypomorphic variants -which have tremendously increased the quality of the manuscript-, strongly support the non-pathogenicity of the VUS. Given the novelty of this approach and the biological complexity of SMA, before formally requesting variant reclassification, our group has initially elected to undertake a complementary two-year transgenic study. Upon completion, we will contact the relevant diagnostic laboratories to initiate formal reclassification procedures.

The discussion and limitation sections have been updated with, for example, the following addition in page 10:

“Our study also strongly supports variant reclassification as benign. Considering the novelty of this approach and the complexity of SMA into late-onset forms, the group decided to continue monitoring the infants and complement our zebrafish rapid assay with long-term transgenic complementation experiments. Formal reclassification will be initiated at the end of this follow-up clinical study and transgenesis approach.”

We believe that this cautious and evidence-based timeline for reclassification does not diminish the significant clinical impact of our approach, which already provides crucial/robust functional evidence to support clinicians in resolving/diagnosing SMA Type I. This is of major impact/benefit for SMA therapy implementation.

Referee #2 (Remarks for Author):

Many important details in the methodology have now been included but there are multiple areas of concern still.

1. In the data provided two data points were excluded in 3B from the smn-/- as 'spurious recordings'. Information on why these were excluded needs to be provided. Further, there is no indication that researchers were blind to the genotype when analysing or excluding data and discussion of blinding, or a statement that it did not occur needs to be included.

The injections were conducted in a randomised sequence - how were they randomised?

As presented below and in line with the reviewer's request, we have performed an entirely new set of experiments conducted in triplicate. The previous datasets are therefore now obsolete. The previously observed “spurious recordings” were linked to the hardware used -the Zebrabox Revolution system-. Occasionally, when closing the device lid, small debris from the insulating black rubber can fall into individual wells of the plate. These artefacts are easily identifiable in the raw recordings, and affected assays are either repeated or, when non-critical for data interpretation, the corresponding wells/source-data are isolated and excluded from the analysis. To prevent confusion, all newly generated Zebrabox experiments presented here were carefully monitored to ensure this issue did not occur in the new source files.

Regarding blinding and randomisation, these information were included in the companion file provided with the EMBO Revision 01. We have now updated and incorporated these details into the Methods section, page 16, as outlined below.

“mRNA injections were performed simultaneously by two independent researchers, each injecting all experimental conditions, and the embryos pooled, except for VUS replicates 1 and 2, which were deliberately restricted to a single operator to assess potential inter-operator variation (Appendix Fig. S2-3). Measurements and analysis were performed without blinding.”

2. Two researchers conducted injections - did they each conduct an independent replicate of each sample? Researcher or replicate has not been considered in the analysis.

Previously, mRNA injections were performed simultaneously by two independent researchers, each injecting all experimental conditions, and the resulting embryos were pooled for analysis. In the current study, all experiments (VUS and hypomorphic variants) were repeated in triplicate -e.g. VUS assays are now presented in six new replicate/dataset-. As before, injections were carried out by two researchers with each injecting all experimental conditions, and embryos were pooled for each replicate, except for VUS replicates 1 and 2, which were deliberately restricted to 1 single operator to assess potential inter-operator variation (Appendix Fig. S2–S3). As shown in the appendix, experimental outcomes were consistent across operators and also replicated across the two sets of triplicates, confirming the reproducibility and robustness of the assay. The methods section on page 16 has been updated to reflect this change.

3. Although the experiment was conducted twice there is no indication as to which data came from which replicate and the lack of a third replicate prevents the consideration of replicate as a factor. In what way were replicates different, what is the level of replication? A third replicate should be completed and power analysis used to determine n-numbers (eg only 6 control fish in fig4B, may not be sufficient given variation).

As briefly stated above, all experiments have now been conducted in triplicate. Each replicate are presented/labelled in Appendix Fig. S2-S4 along with the source files. We have now conducted power calculation to determine the minimum sample size and supplemented the document with the following statement, page 7:

“To confirm the robustness and reproducibility of these findings, the assays were independently repeated three times. As shown in Appendix Fig. S2–S4, all replicates yielded consistent outcomes. Power analysis using the triplicate datasets at 5 dpf revealed a very large effect size (Cohen’s $d \approx 3.3$; ANOVA $f \approx 1.65$, one-way ANOVA, GPower 3.1) between positive and negative controls, indicating that as few as three larvae per group were sufficient to discriminate pathogenic from non-pathogenic variants at 90% power ($\alpha = 0.05$), underscoring the high reliability and sensitivity of this functional assay.”

4. In fig 4 wt mRNA injected animals were present in the survival analysis? Why were these not analysed for swimming?

In response to the reviewer’s question, please note that, given the initial urgency to complete these experiments and provide clinicians with timely results to inform treatment decisions, each VUS was initially assessed independently with a focus on the survival assay. At that stage, due to both time constraints and limited access to the behavioural tracking system, we prioritised inclusion of pathogenic and non-pathogenic controls in the swimming assays rather than the wild-type mRNA condition. During the revision process, however, we successfully extended our analyses to include all experimental conditions, now presented in the revised manuscript.

5. Also in figure 4 there are 12 Tg(SMN1) fish at day 6 and 7 but only 6 at day 5 - why is the data from

the other fish missing or excluded?

As for point 4, the discrepancy in sample numbers resulted from technical limitations during the initial behavioural recordings, as we were constrained by the number of multiwell plates that could be processed on these specific days. No data were excluded. All samples and conditions have now been repeated and fully completed -included in the revised dataset and figures-.

6. The error bars in Figure 4B are not aligned with the graphs or are missing. It is not indicated in any of the figure or legends what the error bars represent.

We thank the reviewer for noting this. The misalignment of the error bars was caused by an export artefact in the previous version of the figure. The newly updated data and associated figures should not be affected by this issue. In addition, the nature of the error bars is now clearly specified in the corresponding figure legends.

7. The additional data in EV1 does provide some encouragement that the conclusion that the variants analysed may not be pathogenic may have been correct. Additional pathogenic controls were requested for the experiments, including those of each mutation class. No additional type I pathogenic controls were included, and the additional variants of the other classes were analysed independently of the VUS. Whilst the results suggest that the assay may be able to detect reduced function in these variants compared to wildtype, it prevented comparison of the two VUS to class II-IV variants. Analysis of swimming was not conducted as for the other variants. This should be included. Furthermore, it appears that the new data was compared to WT control data from a different experiment? The controls should be from the same experiment and therefore this would need to be repeated. Given the degradation of injected RNA over time caution should be used in the interpretation of the survival data at these later timepoints, again increasing the value of the earlier swimming analyses.

The reviewer may not fully appreciate the substantial methodological challenges associated with testing all variants simultaneously, which greatly increases experimental complexity and handling constraints. Nevertheless, to fully address this concern, we have repeated the analyses of all variants (including all controls and both VUS) within the same experimental series and in triplicate. These new experiments also include an additional Type I pathogenic control and now incorporate quantitative motor-function analyses, as requested. Given the large volume of new data generated, the original Figure EV1 has been reorganised into three main figures (Figures 4, 5, and EV1) and two supplementary figures (Appendix Figures S5–S6).

8. In the author's response they state that details of the mock-injections are provided in the reference, but no reference is provided.

It seems the reference has not been entered properly. To avoid confusion, we have added the details of the mix used for our routine mock-injections in the Methods section, page 14: "Mock-injected control embryos were injected with vehicle only (0.1% phenol red in water) and did not receive any mRNA."

9. The new data is provided in the discussion section and should be incorporated into the results section. We have now substantially expanded the experiments for the hypomorphic variants and incorporated these new data into the Results section, positioned immediately before the "Interpretation of findings" section. In accordance with the Editor's request, the manuscript structure has been updated so that the Results section is now titled "Results and Discussion," and the Discussion section has been renamed "Interpretation of findings."

10. In summary there has been some increase in the level of detail necessary and the requested experiments were partially completed, however, there are still significant concerns over lack of sufficient replication, data exclusion, absence, and analysis. Furthermore, the contribution of the zebrafish evidence to

the decision not to treat is not clear, with it having no effect on variant classification.

We hope that our revision has provided the reviewer with the clarity and additional experiments required for full satisfaction.

Regarding the clinical decision not to initiate treatment and the postponement of formal variant reclassification, we have clarified this point in the revised manuscript (page 10):

“Given the pressing need to reach a timely clinical decision, and because i) both VUS proved to be non-pathogenic in the zebrafish assays, ii) the patients were still asymptomatic at six months of age and iii) case 2 carries no SMN2 copy (associated with early pathogenicity or lethality), the clinical team decided not to initiate therapy but instead to continue monitoring until twelve months of age before finalising this report. At that time, both infants were within normal growth parameters and met expected motor milestones. At the time of manuscript submission, both patients were 14 months of age and remained asymptomatic. The initiation of drug therapy with potential side effects and the physical and psychological stress for the child and parents could be avoided. In addition, the public health system did not bear the therapy costs of around US\$2 million (or more) per child.

Our study also strongly supports variant reclassification as benign. Considering the novelty of this approach and the complexity of SMA into late-onset forms, the group decided to continue monitoring the infants and complement our zebrafish rapid assay with long-term transgenic complementation experiments. Formal reclassification will be initiated at the end of this follow-up clinical study and transgenesis approach.”

We believe this clarification now explicitly details the rationale for the clinical decision not to treat, demonstrates the central/essential contribution of the zebrafish results to that decision, and delineates our strategy before initiating formal variant reclassification.

24th Nov 2025

Dear Dr. Giacomotto,

Please find enclosed the final report on your manuscript. We are pleased to inform you that your manuscript is accepted for publication.

Given the remaining comment from Reviewer #1, we would still ask you to use a more cautious wording regarding the statement there is no functional SMN present from the maternal egg. You can send us the revised manuscript by replying to this email and we will upload it on our end.

After that has been done, your paper will be sent to our publisher to be included in the next available issue of EMBO Molecular Medicine. Your manuscript will then be processed for publication by EMBO Press. It will be copy edited and you will receive page proofs prior to publication. Please note that you will be contacted by Springer Nature Author Services to complete licensing and payment information.

Sincerely,
Jingyi

Jingyi Hou
Senior Editor
EMBO Molecular Medicine

Referee #1 (Comments on Novelty/Model System for Author):

The model system is clear and assays the alleles in the correct manner the identification of a non pathogenic variant allows the prevention of unneeded treatment

Referee #1 (Remarks for Author):

The use of the maternal zygotic fish is excellent I would just say caution should be used in describing it as fully deprived of SMN function. The Mz fish are based on a Smn expression construct under the heat shock promoter and to a very low degree this is leaky, so the possibility exists that the maternal egg has a very low level of SMN. To be honest this is as low as you can possibly get. The point being SMN is essential for cells to divide for sure the effects seen at 3 days - 5 days are severe and death occurs very quickly. I really feel the use of the fish in this situation is the best it can be. A completely non-functional allele will basically do nothing as shown by the authors. A mild allele is being assayed in the presence of a very small amount of SMN which allows complementation and therefore some improvement. This is perfect because it mimics how mild SMA alleles work the A2G mutant is modeled here but that allele is non-functional on a mouse Smn^{-/-} background but functional on a SMN2 containing mouse. The authors do discuss this in limitations as well as the difficulty of Smn detection by western at these very early stages. The only other way to address it would be to grow a cell line out of the very early zebrafish development time point a bit like an ES cell, but this is way beyond what is required for the current paper. The point is that the authors have clearly shown an excellent system to model mutants or VUS as to whether they are functional or not as such the paper provides excellent information. I would just encourage a more refrained wording I know of no cell from anything that can function and divide with no Smn.

The author has addressed all previous criticisms including completely re-doing the assays for Vus and mutants with the required number of replicates. In some ways the previous discussion in review has some flaws in that there are modifiers that lie outside the SMN locus the view of this reviewer is these are as yet to be clearly identified and as such the difference between a type II variant and a type III or IV variant is questionable at best as you have no idea whether the modifier located elsewhere in the genome is present or not and this is likely to influence the exact phenotype. So basically, the authors have extracted the

information possible from the system and convincingly shown the two VUS mutation in the C terminal to be functional. The assay also gives good information rapidly on mild mutations.

The only other point I would raise is that if a person is born with only one SMN gene with a mutation that has to have at least some function as loss of SMN completely does not give rise to a viable fetus even with one copy of SMN2 the phenotype would be predicted to be a type 0 and present at birth if the allele was not functional. So, the assay development and use is great and the only thing I would request is not to be so adamant that there is no functional SMN present from the maternal egg. I agree with the authors on holding off on completely changing the VUS to begin but currently it probably should be classified as likely begin but this is really not critical to the importance of the paper and the changing from a VUS to a begin variant will be bureaucratic.

Referee #1 (Comments on Novelty/Model System for Author):

The model system is clear and assays the alleles in the correct manner the identification of a non pathogenic variant allows the prevention of unneeded treatment

Referee #1 (Remarks for Author):

The use of the maternal zygotic fish is excellent I would just say caution should be used in describing it as fully deprived of SMN function. The Mz fish are based on a Smn expression construct under the heat shock promoter and to a very low degree this is leaky, so the possibility exists that the maternal egg has a very low level of SMN. To be honest this is as low as you can possibly get. The point being SMN is essential for cells to divide for sure the effects seen at 3 days - 5days are severe and death occurs very quickly. I really feel the use of the fish in this situation is the best it can be. A completely non- functional allele will basically do nothing as shown by the authors. A mild allele is being assayed in the presence of a very small amount of SMN which allows complementation and therefore some improvement. This is perfect because it mimics how mild SMA alleles work the A2G mutant is modeled here but that allele is non- functional on a mouse Smn^{-/-} background but functional on a SMN2 containing mouse. The authors do discuss this in limitations as well as the difficulty of Smn detection by western at these very early stages. The only other way to address it would be to grow a cell line out of the very early zebrafish development time point a bit like an ES cell, but this is way beyond what is required for the current paper. The point is that the authors have clearly shown an excellent system to model mutants or VUS as to whether they are functional or not as such the paper provides excellent information. I would just encourage a more refrained wording I know of no cell from anything that can function and divide with no Smn.

The author has addressed all previous criticisms including completely re-doing the assays for Vus and mutants with the required number of replicates. In some ways the previous discussion in review has some flaws in that there are modifiers that lie outside the SMN locus the view of this reviewer is these are as yet to be clearly identified and as such the difference between a type II variant and a type III or IV variant is questionable at best as you have no idea whether the modifier located elsewhere in the genome is present or not and this is likely to influence the exact phenotype. So basically, the authors have extracted the information possible from the system and convincingly shown the two VUS mutation in the C terminal to be functional. The assay also gives good information rapidly on mild mutations.

The only other point I would raise is that if a person is born with only one SMN gene with a mutation that has to have at least some function as loss of SMN completely does not give rise to a viable fetus even with one copy of SMN2 the phenotype would be predicted to be a type 0 and present at birth if the allele was not functional. So, the assay development and use is great and the only thing I would request is not to be so adamant that there is no functional SMN present from the maternal egg. I agree with the authors on holding off on completely changing the VUS to begin but currently it probably should be classified as likely begin but this is really not critical to the importance of the paper and the changing from a VUS to a begin variant will be bureaucratic.
